# Provably Learning Attention with Queries

**Satwik Bhattamishra** [1]  **Kulin Shah** [2]  **Michael Hahn** [3]  **Varun Kanade** [1]

## Abstract

We study the problem of learning Transformer-based sequence models with black-box access to their outputs. In this setting, a learner may adaptively query the oracle with any sequence of vectors and observe the output of the target function. We begin with studying the learnability of the simplest formulation, that is, learning a single-head attention-based regressor with queries. We show that for a model with width $d$, there is an elementary algorithm to learn the parameters of single-head attention with $O(d^2)$ queries. Further, we show that if there exists an algorithm to learn ReLU feedforward networks (FFNs), then the single-head algorithm can be easily adapted to learn one-layer Transformers with single-head attention. Next, we show that, in the common regime where the head dimension $r \ll d$, single-head attention-based models can be learned with $O(rd)$ queries via compressed sensing arguments. We also study robustness to noisy oracle access, proving that under mild norm and margin conditions, the parameters can be estimated to $\varepsilon$ accuracy with a polynomial number of queries even when outputs are only provided up to additive tolerance. Finally, we consider the learnability of multi-head attention and show that they are not identifiable from queries, and hence, learnability in the same sense is not feasible without additional assumptions. We discuss potential approaches to learn multi-head attention-based models under certain structural assumptions.

## 1. Introduction

Transformer-based models (Vaswani et al., 2017) are being widely deployed in the industry. Given their ubiquity, a natural question is the following: given black-box access

[1] University of Oxford [2] UT Austin [3] Saarland University. Correspondence to: Satwik Bhattamishra <satwik.bmishra@cs.ox.ac.uk or satwik55@gmail.com>.

*Proceedings of the 43rd International Conference on Machine Learning*, Seoul, South Korea. PMLR 306, 2026. Copyright 2026 by the author(s).

to the outputs of a target model (e.g., via an API), can an adversary recover the weights of the target model? Such questions regarding model stealing have been widely studied empirically and theoretically for feedforward and related networks (Tramèr et al., 2016; Shi et al., 2017; Jagielski et al., 2020; Orekondy et al., 2019).

A natural formalisation of this problem is the problem of *learning with queries* (Bshouty, 2013; Angluin, 1988) where a learner adaptively chooses inputs and observes labels (or real-valued outputs) with the goal of reconstructing the target function or, more stringently, the target parameters. Beyond security-driven model extraction, parameter recovery is also a natural lens for understanding what aspects of a model are identifiable from behaviour alone. A strand of work has focused on providing provable guarantees for recovering ReLU networks under value or membership queries, often under additional structural assumptions (Chen et al., 2021; Milli et al., 2019; Daniely & Granot, 2023). However, parameter recovery guarantees for softmax-attention from access to queries have been largely underexplored.

**Problem.** We study parameter recovery for attention-based sequence models with *value-query* access. Concretely, we begin with a single-head attention-based regressor which can be formulated in the following way (see Sec. 3 for details). For an input sequence $X \in \mathbb{R}^{N \times d}$ (with variable length $N \geq 1$) and parameters $W \in \mathbb{R}^{d \times d}$ and $v \in \mathbb{R}^d$, the model computes $f_{W,v}(X) \in \mathbb{R}$ where

$$f_{W,v}(X) = \text{softmax}\big(x_1^\top W x_N, \ldots, x_N^\top W x_N\big)^\top (Xv).$$

The last token plays the role of the query token in the attention mechanism and the softmax produces attention weights over positions. A learner receives black-box access to $\text{VQ}(X) = f_{W^\star,v^\star}(X)$ and can query *any* sequence $X$; the goal is to recover $(W^\star, v^\star)$ exactly or approximately.

**Contributions.** We initiate a systematic study of learning softmax attention with value queries and provide the first, to our knowledge, provable guarantees for parameter recovery. We analyse learnability in different query models and regimes, and establish the following results.

**(i) Learning single-head attention.** We give an elementary, polynomial-time algorithm that exactly recovers $(W^\star, v^\star)$ for the single-head attention regressor using $O(d^2)$ value

queries (Theorem 4.1). The key observation is that we can use short sequence vectors to isolate the softmax nonlinearity. Using queries of length two reduces softmax to a sigmoid, which can be inverted, turning each oracle response into linear equations that can be solved to obtain the target parameters. Further, we show that the single-head algorithm can be lifted to learn a one-layer, single-head Transformer that composes attention with a two-layer ReLU MLP: assuming an algorithm exists for learning the corresponding ReLU FFN from value queries, we show it can be combined with our attention-recovery method to obtain a value-query learner for one-layer Transformer (Section 4.1).

**(ii) Faster recovery in low-rank regime.** Motivated by the common setting where the head dimension $r \ll d$, we analyse the case $\text{rank}(W^\star) \leq r$. We give a randomized algorithm that recovers $(W^\star, v^\star)$ using $O(rd)$ queries (Theorem 5.1). The main idea is to design probes that implement rank-one linear measurements of $W^\star$, reducing recovery to a standard low-rank matrix sensing problem and enabling reconstruction via compressed sensing tools.

**(iii) Robustness and Stability.** Our exact recovery algorithm relies on inverting a sigmoid, which is not globally Lipschitz. We therefore study a more realistic oracle that returns values within additive tolerance $\tau$. Under mild norm bounds and a simple margin condition, we show that one can recover $(W^\star, v^\star)$ to $\epsilon$ accuracy in polynomial time with $O(d^2)$ queries and with tolerances scaling polynomially in $d$ and the target accuracy (Section 6). The proof constructs probes that keep the recovered attention weights in a stable range where the inverse-sigmoid behaves well.

We also extend our approach to the weaker membership-query setting, where the oracle returns only a binary label instead of real-valued scalars. Since direct sigmoid inversion is no longer available, we combine the stability-based probe design with a bisection method to approximately recover $(W^\star, v^\star)$ under slightly stronger assumptions (App. B).

Lastly, we show that multi-head attention parameters are *not identifiable* from value queries in general: distinct parameter settings can induce exactly the same input–output map (Proposition 7.1). Consequently, guarantees analogous to the single-head case cannot hold without additional assumptions. We discuss some structural conditions that restore identifiability and outline possible query-based approaches in this regime (App. C).

At a high level, our first main result shows that single-head softmax attention admits surprisingly straightforward parameter recovery. This stands in contrast to learning with random examples (without queries), where the problem is non-convex for standard loss functions and is quite challenging. Further, the analogous problem of learning a one-hidden-layer ReLU MLP with queries, which is similar in

surface form, has also been substantially more challenging (Chen et al., 2021).

## 2. Related Work

**Learning Neural Nets with Queries.** Learning with membership/value queries is a classical topic in computational learning theory (Angluin, 1988; Angluin et al., 1993), and it is also a natural formalisation of black-box *model extraction* where an adversary adaptively probes a prediction API. Early work investigated whether a feedforward network can be reconstructed from oracle access to its input–output map (Fefferman & Markel, 1993). More recently, empirical methods have been proposed for extraction attacks and have been studied extensively in the context of security and ML (Tramèr et al., 2016; Orekondy et al., 2019; Jagielski et al., 2020). On the theory side, there are now polynomial-time guarantees for learning shallow ReLU networks under different query models and structural assumptions, including value-query learning under Gaussian inputs (Chen et al., 2021), linear independence of parameter vectors (Milli et al., 2019), and exact parameter extraction under general-position conditions (Daniely & Granot, 2023). In contrast, despite the central role of attention in modern sequence models, we are not aware of prior work giving provable learnability/parameter estimation guarantees for softmax attention (or Transformers built from it) from value queries alone; our results initiate this study for attention blocks. Complementary to our work, Carlini et al. (2024); Finlayson et al. (2024) tackle the problem of extracting specific parts of a production-level language model, where they recover limited information such as width and embedding matrices, but in a more challenging and realistic setting.

**Learning Single-layer Attention.** There is a growing body of work that studies the passive learnability of a single-layer attention-based model from examples under various assumptions (e.g. Chen & Li, 2025; Tian et al., 2023; Magen et al., 2024). For instance, Wang et al. (2024) analyse the learnability of sparse token selection with a single-head attention model and prove separations from fully-connected nets. On the optimisation side, Arnaboldi et al. (2025) study training dynamics for simplified attention-style models, and recent high-dimensional/statistical-mechanics analyses quantify when softmax attention detects weak sparse signals and how it compares to linear alternatives (Marion et al., 2025; Barnfield et al., 2025). Complementarily, several works characterise implicit bias and geometry in attention training, including max-margin/token-selection viewpoints and connections to SVMs (Ataee Tarzanagh et al., 2023; Tarzanagh et al., 2023; Sheen et al., 2024), and Huang et al. (2024) study learning dynamics for in-context learning with one-layer softmax Transformers. Finally, when softmax is removed (linear attention), Yau et al. (2024) show PAC

learnability via a reduction to kernel methods. These works focus on passive learning and optimisation from random examples, whereas our work studies efficient learnability and parameter recovery with queries.

## 3. Definitions

Each input example consists of a matrix–label pair $(X, y)$ where $X \in \mathbb{R}^{N \times d}$ contains $N$ row-vectors $X = \begin{pmatrix} \boldsymbol{x}_1^\top \\ \vdots \\ \boldsymbol{x}_N^\top \end{pmatrix}$, $\boldsymbol{x}_i \in \mathbb{R}^d$, and the target label is a scalar $y \in \mathbb{R}$.

**Model class** ATT. Fix an integer $d \geq 1$ and the model can take inputs of any length $N \geq 1$. We focus on a single attention-based regressor that takes a sequence of vectors as input and produces a scalar as output. Let $\boldsymbol{Q}, \boldsymbol{K}, \boldsymbol{V} \in \mathbb{R}^{r \times d}$ be the query, key, and value projection matrices. Let $\boldsymbol{w}_o \in \mathbb{R}^r$ be the output projection vector. For any $N$, the output of a single attention-based model $\mathrm{SHA} : \mathbb{R}^{N \times d} \to \mathbb{R}$ is computed by $\mathrm{SHA}(X) =$

$$\mathrm{softmax}(\boldsymbol{x}_1^\top \boldsymbol{K}^\top \boldsymbol{Q} \boldsymbol{x}_N, \ldots, \boldsymbol{x}_N^\top \boldsymbol{K}^\top \boldsymbol{Q} \boldsymbol{x}_N)^\top (X \boldsymbol{V}^\top \boldsymbol{w}_o)$$

The model can be viewed in a simpler way in which the query and key projections are merged into one matrix, i.e., $\boldsymbol{x}_i \boldsymbol{K}^\top \boldsymbol{Q} \boldsymbol{x}_N = \boldsymbol{x}_i \boldsymbol{W} \boldsymbol{x}_N$ for some matrix $\boldsymbol{W} \in \mathbb{R}^{d \times d}$. Additionally, the value projection and output vector can be merged: $X \boldsymbol{V}^\top \boldsymbol{w}_o = X \boldsymbol{v}$ for some vector $\boldsymbol{v} \in \mathbb{R}^d$. More precisely, an equivalent definition is as follows. For parameters $\boldsymbol{W} \in \mathbb{R}^{d \times d}$ and $\boldsymbol{v} \in \mathbb{R}^d$ define the score vector $s_i(X, \boldsymbol{W}) = \boldsymbol{x}_i^\top \boldsymbol{W} \boldsymbol{x}_N$, for $i = 1, \ldots, N$, and the (row) attention weights

$$\boldsymbol{\alpha}(X, \boldsymbol{W}) = \mathrm{softmax}(s_1, \ldots, s_N) \in \Delta^{N-1},$$

where $\mathrm{softmax}(z)_i = \exp(z_i) / \sum_j \exp(z_j)$ and $\Delta^{N-1}$ is the probability simplex. The attention model associated with $(\boldsymbol{W}, \boldsymbol{v})$ is the map

$$f_{\boldsymbol{W}, \boldsymbol{v}}(X) = \boldsymbol{\alpha}(X, \boldsymbol{W})^\top (X \boldsymbol{v}) \in \mathbb{R}.$$

With all such functions, we have the hypothesis class

$$\mathrm{ATT}_d = \left\{ f_{\boldsymbol{W}, \boldsymbol{v}} \,\middle|\, \boldsymbol{W} \in \mathbb{R}^{d \times d} \text{ and } \boldsymbol{v} \in \mathbb{R}^d \right\}.$$

The class $\mathrm{ATT} = \cup_{d \geq 1} \mathrm{ATT}_d$. The single-head attention-based model considered here is essentially a one-layer Transformer with a linear network instead of a ReLU MLP.

**Learning objective.** Throughout, we primarily focus on parameter estimation from value or membership queries. By *learning*, we will mean exact or approximate identification of the target parameters. Given black-box access to the target function, the goal of the learner is to identify the target either exactly or approximately by querying the

target function. For instance, for a target function $f^\star$ parameterised by $(\boldsymbol{W}^\star, \boldsymbol{v}^\star)$, the learner is given access to a value query oracle $\mathrm{VQ}(X) = f^\star(X)$ which returns the true output exactly (or in some cases approximately as in Sec. 6). The learner can query any input $X \in \mathcal{X}$ to the value query oracle and the learner aims to recover $(\widehat{\boldsymbol{W}}, \widehat{\boldsymbol{v}})$ such that $(\widehat{\boldsymbol{W}}, \widehat{\boldsymbol{v}}) = (\boldsymbol{W}^\star, \boldsymbol{v}^\star)$, or for some $\epsilon \in \mathbb{R}$,

$$\|\boldsymbol{W}^\star - \widehat{\boldsymbol{W}}\|_F \leq \epsilon \quad \text{and} \quad \|\boldsymbol{v}^\star - \widehat{\boldsymbol{v}}\|_2 \leq \epsilon.$$

A class of functions is efficiently learnable if the number of queries used and the runtime of the learning algorithm are polynomial in the parameters of the concept class (such as $d$ for the class ATT).

## 4. Learning Attention with Queries

We will analyse whether an unknown single-head attention-based regressor can be learned exactly using value queries only. The learner has access to the value query oracle $f_{\boldsymbol{W}^\star, \boldsymbol{v}^\star} : \mathbb{R}^{N \times d} \to \mathbb{R}$ such that for any $X \in \mathbb{R}^{N \times d}$, it can query the label $f_{\boldsymbol{W}^\star, \boldsymbol{v}^\star}(X) \in \mathbb{R}$. We describe a value query (VQ) method that exactly identifies the target parameters $(\boldsymbol{W}^\star, \boldsymbol{v}^\star)$ whenever $\boldsymbol{v}^\star \neq \boldsymbol{0}$. Throughout, let $\mathbf{e}_i \in \mathbb{R}^d$ denote the $i$th standard basis vector, and let $\sigma(t) = 1/(1 + e^{-t})$ denote the sigmoid function.

**Theorem 4.1.** *Assuming $\boldsymbol{v}^\star \neq \boldsymbol{0}$, given access to a value query oracle $\mathrm{VQ}(X) = f_{\boldsymbol{W}^\star, \boldsymbol{v}^\star}(X)$, the parameters $\boldsymbol{W}^\star$ and $\boldsymbol{v}^\star$ are exactly learnable in polynomial-time with $O(d^2)$ value queries.*

*Proof.* The algorithm has two phases. The first phase is trivial, and recovers $\boldsymbol{v}^\star$ using one-row or length-one queries. The second phase recovers $\boldsymbol{W}^\star$ column–by–column using two-row or length-two queries.

*(i) Identifying $\boldsymbol{v}^\star$.* With $N = 1$ the softmax weight is 1, so $f_{\boldsymbol{W}, \boldsymbol{v}}([\boldsymbol{x}^\top]) = \boldsymbol{x}^\top \boldsymbol{v}$ and is independent of $\boldsymbol{W}$. Query the $d$ one–row inputs $X = [\mathbf{e}_i^\top]$ for $i = 1, \ldots, d$ to obtain $y = \boldsymbol{v}_i^\star$. Hence, the vector $\boldsymbol{v}^\star$ is recovered with $d$ VQs.

*(ii) Identifying $\boldsymbol{W}^\star$ (column–wise recovery).* Fix $j \in \{1, \ldots, d\}$ and let $\mathbf{w}_j := \boldsymbol{W}^\star \mathbf{e}_j \in \mathbb{R}^d$ and $(\mathbf{w}_j)_i = \boldsymbol{W}_{ij}^\star$. For any probe vector $\boldsymbol{u} \in \mathbb{R}^d$, consider the two–row input

$$X = \begin{bmatrix} (\boldsymbol{u} + \mathbf{e}_j)^\top \\ \mathbf{e}_j^\top \end{bmatrix}.$$

The attention scores are

$$s_1 = (\boldsymbol{u} + \mathbf{e}_j)^\top \boldsymbol{W}^\star \mathbf{e}_j = (\boldsymbol{u} + \mathbf{e}_j)^\top \mathbf{w}_j,$$

$$s_2 = \mathbf{e}_j^\top \boldsymbol{W}^\star \mathbf{e}_j = (\mathbf{w}_j)_j.$$

So the attention weight on the first row or position is

$$\alpha(\boldsymbol{u}; j) = \frac{e^{s_1}}{e^{s_1} + e^{s_2}} = \sigma(s_1 - s_2) = \sigma(\boldsymbol{u}^\top \mathbf{w}_j), \quad (1)$$

where $\sigma(t) = 1/(1 + e^{-t})$. Since $(X\boldsymbol{v}^\star) = \begin{bmatrix} \boldsymbol{u}^\top \boldsymbol{v}^\star + \boldsymbol{v}_j^\star, \boldsymbol{v}_j^\star \end{bmatrix}^\top$, the label returned by the oracle is the convex combination

$$y = \alpha(\boldsymbol{u}; j)\left((\boldsymbol{u} + \mathbf{e}_j)^\top \boldsymbol{v}^\star\right) + \left(1 - \alpha(\boldsymbol{u}; j)\right)\boldsymbol{v}_j^\star$$

$$= \boldsymbol{v}_j^\star + \alpha(\boldsymbol{u}; j)\left(\boldsymbol{u}^\top \boldsymbol{v}^\star\right).$$

If $\boldsymbol{u}^\top \boldsymbol{v}^\star \neq \boldsymbol{0}$, we can recover the attention weight from the observed $y$ via

$$\alpha(\boldsymbol{u}; j) = \frac{y - \boldsymbol{v}_j^\star}{\boldsymbol{u}^\top \boldsymbol{v}^\star} \in (0, 1).$$

Thus, combining with (1), we have,

$$\boldsymbol{u}^\top \mathbf{w}_j = \sigma^{-1}\left(\frac{y - \boldsymbol{v}_j^\star}{\boldsymbol{u}^\top \boldsymbol{v}^\star}\right)$$

Thus, each probe vector $\boldsymbol{u}$ yields a *linear* equation in the unknown column $\mathbf{w}_j$. This leads to one linear equation for $d$ variables in $\mathbf{w}_j$. We can use $d$ linearly independent vectors to identify $\mathbf{w}_j$ by solving a system of linear equations.

Let $\boldsymbol{u}_1, \ldots, \boldsymbol{u}_d$ be $d$ linearly independent vectors such that $\boldsymbol{u}_\ell^\top, \boldsymbol{v}^\star \neq \boldsymbol{0}$ for all $\ell \in [d]$. Take invertible $\boldsymbol{Z} = [\boldsymbol{u}_1^\top; \ldots; \boldsymbol{u}_d^\top] \in \mathbb{R}^{d \times d}$ and collect $d$ measurements $\boldsymbol{t} = [t_1, \ldots, t_d]$ with $\boldsymbol{Z}\mathbf{w}_j = \boldsymbol{t}$. One can recover the $j$th column of $\boldsymbol{W}^\star$ with $\mathbf{w}_j = \boldsymbol{Z}^{-1}\boldsymbol{t}$ and repeating this for $j = 1, \ldots, d$ recover $\boldsymbol{W}^\star$.

One could sample $d$ vectors $\boldsymbol{u}_1, \ldots, \boldsymbol{u}_d$ i.i.d. from $\mathcal{N}(0, I_d)$ and they will be linearly independent and the inner product $\boldsymbol{u}_\ell^\top \boldsymbol{v}^\star \neq 0$ for all $\ell$ almost surely (with probability 1). Another intuitive approach to understand how the parameters are recovered is the following. Pick any index $p$ with $\boldsymbol{v}_p^\star \neq 0$. For $\ell = 1, \ldots, d$, set the probe vector $\boldsymbol{u}_\ell$ in the following way,

$$\boldsymbol{u}_\ell = \begin{cases} \mathbf{e}_\ell, & \text{if } \boldsymbol{v}_\ell^\star \neq 0, \\ \mathbf{e}_\ell + \mathbf{e}_p, & \text{if } \boldsymbol{v}_\ell^\star = 0. \end{cases}$$

For any column vector $\mathbf{w}_j$, it is easy to see that the measurements $t_\ell = \boldsymbol{u}_\ell^\top \mathbf{w}_j = \boldsymbol{W}_{\ell j}^\star$ when $\boldsymbol{v}_\ell^\star \neq 0$ (e.g. for $\ell = p$) and $t_\ell = \boldsymbol{W}_{\ell j}^\star + \boldsymbol{W}_{p j}^\star$ when $\boldsymbol{v}_\ell^\star = 0$.

The first phase uses $d$ queries to recover $\boldsymbol{v}^\star$ and the second phase uses $d$ VQs per column, totalling $d^2$ additional queries. Hence, the entire procedure uses $O(d^2)$ VQs and runs in polynomial time. If $\boldsymbol{v}^\star = \boldsymbol{0}$, then $f_{\boldsymbol{W}^\star, \boldsymbol{v}^\star} \equiv 0$ for all $X$ and $\boldsymbol{W}^\star$ is not identifiable from VQs (many $\boldsymbol{W}$ induce the same function). Conversely, when $\boldsymbol{v}^\star \neq \boldsymbol{0}$, the column–wise linear system above uniquely determines $(\boldsymbol{W}^\star, \boldsymbol{v}^\star)$.

$\square$

**Discussion.** Our result indicates that the problem of learning a single-head attention model with value queries is relatively

straightforward. It is worth discussing why this could be somewhat surprising. First, in contrast, proving comparable guarantees from i.i.d. random examples (i.e., without queries) for attention-style models typically requires nontrivial assumptions on the data model, task, or training dynamics, and substantial effort has gone into understanding learnability and establishing such results even in restricted settings (Wang et al., 2024; Tian et al., 2023; Arnaboldi et al., 2025; Marion et al., 2025; Magen et al., 2024). Similar to the problem of learning a single-head attention model, the problem of learning a one-hidden-layer ReLU MLP also has a single nonlinearity together with a parameter matrix and a vector. However, this analogous problem appears substantially more challenging: under minimal assumptions, existing efficient guarantees are generally distribution-dependent and focus on approximate function recovery (e.g., under Gaussian marginals), while exact recovery typically requires additional structural assumptions; see Chen et al. (2021) for further discussion. Lastly, it is worth noting that many elementary concept classes, such as conjunctions or singletons, are not efficiently learnable with access to queries alone.

### 4.1. Learning 1-Layer Transformers

In this section, we show how to learn a one-layer Transformer with single-head attention, assuming a value-query learner for two-layer ReLU MLPs exists. In particular, given an algorithm $\mathcal{A}_{\text{FFN}}$ that learns two-layer ReLU MLPs from value queries, we give a method that combines $\mathcal{A}_{\text{FFN}}$ with our algorithm for single-head attention to learn a one-layer, single-head Transformer.

**Model and Setup.** A one-layer Transformer with single-head attention is the composition of a ReLU MLP and a single-head attention block. The attention block produces a single vector (here we attend using the last token as the query), and the FFN maps $\mathbb{R}^d \to \mathbb{R}$. Concretely, let $\boldsymbol{Q}, \boldsymbol{K} \in \mathbb{R}^{r \times d}$ and $\boldsymbol{V} \in \mathbb{R}^{d \times d}$ denote the query, key, and value projections, and let the FFN be a two-layer ReLU network $\text{FFN}(z) = \boldsymbol{w}_2^\top \text{ReLU}(\boldsymbol{W}_1 z)$ with $\boldsymbol{W}_1 \in \mathbb{R}^{m \times d}$ and $\boldsymbol{w}_2 \in \mathbb{R}^m$. We assume the FFN has no bias terms. Then for $X \in \mathbb{R}^{N \times d}$ the model can be written as $\text{TF}(X) =$

$$\boldsymbol{w}_2^\top \text{ReLU}\Big(\boldsymbol{W}_1 \cdot \big[\text{softmax}(\boldsymbol{x}_i^\top \boldsymbol{K}^\top \boldsymbol{Q}\boldsymbol{x}_N)^\top (X\boldsymbol{V}^\top)\big]\Big).$$

As in Section 3, we merge query and key parameters by setting $\boldsymbol{W} := \boldsymbol{K}^\top \boldsymbol{Q} \in \mathbb{R}^{d \times d}$, so the scores are $s_i(X, \boldsymbol{W}) = \boldsymbol{x}_i^\top \boldsymbol{W} \boldsymbol{x}_N$ and $\boldsymbol{\alpha}(X, \boldsymbol{W}) = \text{softmax}(s_1, \ldots, s_N)$. We also merge the value map and the first FFN layer by defining $\boldsymbol{A} := \boldsymbol{V}^\top \boldsymbol{W}_1^\top \in \mathbb{R}^{d \times m}$, and rename the second-layer vector $\boldsymbol{w}_o := \boldsymbol{w}_2$. With these reparameterizations,

$$\text{TF}(X) = \boldsymbol{w}_o^\top \text{ReLU}\big(\boldsymbol{\alpha}(X, \boldsymbol{W})^\top X \boldsymbol{A}\big),$$

which is the same architecture with the three parameters

$(\boldsymbol{W}, \boldsymbol{A}, \boldsymbol{w}_o)$. We assume access to a value-query oracle $\mathrm{VQ}(X) = \mathrm{TF}_{\boldsymbol{W}^\star, \boldsymbol{A}^\star, \boldsymbol{w}_o^\star}(X)$.

**Using an FFN learner.** We will reduce learning TF to (i) learning a two-layer ReLU network from black-box queries and (ii) learning single-head attention as in Theorem 4.1. Formally, assume there exists an algorithm $\mathcal{A}_{\mathrm{FFN}}$ that, given oracle access to a function of the form $x \mapsto \boldsymbol{w}_o^{\star\top} \mathrm{ReLU}(x^\top \boldsymbol{A}^\star)$, outputs parameters $(\widehat{\boldsymbol{A}}, \widehat{\boldsymbol{w}}_o)$ computing the *same* function on all $x \in \mathbb{R}^d$. To invoke $\mathcal{A}_{\mathrm{FFN}}$, we use one-row queries. For $X = [x^\top]$ we have the scalar identity $\mathrm{VQ}([x^\top]) = \boldsymbol{w}_o^{\star\top} \mathrm{ReLU}(x^\top \boldsymbol{A}^\star)$. Therefore, restricting the Transformer oracle to length-1 inputs results in exactly the FFN target needed by $\mathcal{A}_{\mathrm{FFN}}$, and we can recover an equivalent pair $(\widehat{\boldsymbol{A}}, \widehat{\boldsymbol{w}}_o)$.

**Recovering parameters of attention.** We next show that $\boldsymbol{W}^\star$ can be recovered independently of $\mathcal{A}_{\mathrm{FFN}}$ by eliminating the ReLU using two queries. Define $\widetilde{\mathrm{VQ}}(X) := \mathrm{VQ}(X) - \mathrm{VQ}(-X)$. For softmax attention, negating all tokens does not change the scores because $s_i(-X, \boldsymbol{W}) = (-\boldsymbol{x}_i)^\top \boldsymbol{W}(-\boldsymbol{x}_N) = s_i(X, \boldsymbol{W})$, and thus $\boldsymbol{\alpha}(-X, \boldsymbol{W}) = \boldsymbol{\alpha}(X, \boldsymbol{W})$. Consequently,

$$\boldsymbol{\alpha}(-X, \boldsymbol{W})^\top (-X)\boldsymbol{A} = -\boldsymbol{\alpha}(X, \boldsymbol{W})^\top X \boldsymbol{A}.$$

Using $\mathrm{ReLU}(u) - \mathrm{ReLU}(-u) = u$ coordinate-wise, we obtain

$$\widetilde{\mathrm{VQ}}(X) = \boldsymbol{w}_o^{\star\top}\Big(\mathrm{ReLU}(z) - \mathrm{ReLU}(-z)\Big) = \boldsymbol{w}_o^{\star\top} z$$

$$= \boldsymbol{\alpha}(X, \boldsymbol{W}^\star)^\top X \boldsymbol{v}^\star, \qquad z = \boldsymbol{\alpha}(X, \boldsymbol{W}^\star)^\top X \boldsymbol{A}^\star,$$

where $\boldsymbol{v}^\star := \boldsymbol{A}^\star \boldsymbol{w}_o^\star \in \mathbb{R}^d$. Thus $\widetilde{\mathrm{VQ}}$ is *exactly* a value-query oracle for the single-head attention regressor $f_{\boldsymbol{W}^\star, \boldsymbol{v}^\star}(X) = \boldsymbol{\alpha}(X, \boldsymbol{W}^\star)^\top (X\boldsymbol{v}^\star)$ considered in Section 4.1. Whenever $\boldsymbol{v}^\star \neq \boldsymbol{0}$, Theorem 4.1 applied to $\widetilde{\mathrm{VQ}}$ recovers $\boldsymbol{W}^\star$ (and $\boldsymbol{v}^\star$) exactly; each query to $\widetilde{\mathrm{VQ}}$ uses two regular value queries.

Together, the two steps lead to a learning algorithm for one-layer single-head Transformers: use length-1 queries and $\mathcal{A}_{\mathrm{FFN}}$ to recover an equivalent FFN $(\widehat{\boldsymbol{A}}, \widehat{\boldsymbol{w}}_o)$, and use antisymmetric queries $\widetilde{\mathrm{VQ}}$ together with Theorem 4.1 to recover $\widehat{\boldsymbol{W}} = \boldsymbol{W}^\star$. If $\boldsymbol{v}^\star = \boldsymbol{A}^\star \boldsymbol{w}_o^\star = \boldsymbol{0}$, then $\widetilde{\mathrm{VQ}} \equiv 0$ and $\boldsymbol{W}^\star$ is not identifiable from such queries. The overall query complexity is $Q_{\mathrm{FFN}}(d, m) + O(d^2)$ up to constant factors, where $Q_{\mathrm{FFN}}(d, m)$ is the number of queries used by $\mathcal{A}_{\mathrm{FFN}}$.

**Discussion.** Several prior works have studied the learnability of ReLU FFNs with queries. Milli et al. (2019) show that when the columns of $\boldsymbol{A}$ are linearly independent, then a ReLU network is efficiently learnable with queries and their algorithm can act as $\mathcal{A}_{\mathrm{FFN}}$ to learn one-layer Transformers under the somewhat strong linear independence assumption. While learning 2-layer ReLU MLPs without any structural assumptions is quite difficult (Chen et al., 2021), Daniely

& Granot (2023) provide an efficient algorithm for learning ReLU FFNs with relatively much milder assumptions. Their algorithm is not directly applicable to FFNs without bias terms, but we conjecture that with some modifications, their algorithm can be applied in this scenario to learn one-layer Transformers.

# 5. Low Rank Recovery

We now consider the problem where the target matrix $\boldsymbol{W}^\star$ has a known upper bound on the rank: $\mathrm{rank}(\boldsymbol{W}^\star) \leq r$. This studies the scenario where the head dimension in an attention block is much smaller than the embedding dimension. In modern usage of Transformers, the head dimension is often small (e.g., 128) relative to the width (e.g., 4096 or 8192) (Touvron et al., 2023; Brown et al., 2020). For query and key projection matrices $\boldsymbol{Q}, \boldsymbol{K} \in \mathbb{R}^{r \times d}$, if $r \ll d$, then $\mathrm{rank}(\boldsymbol{W}^\star) = \mathrm{rank}(\boldsymbol{K}^\top \boldsymbol{Q}) \leq r$. We show that in such a scenario, we can leverage techniques from compressed sensing (Cai & Zhang, 2015) to recover the target parameters $\boldsymbol{W}^\star, \boldsymbol{v}^\star$ with $O(rd)$ queries, which is much more efficient than $O(d^2)$.

In simple terms, Cai & Zhang (2015) show that for an unknown matrix $\boldsymbol{W} \in \mathbb{R}^{d \times d}$ with $\mathrm{rank}(\boldsymbol{W}) \leq r$, if one can obtain $O(rd)$ rank-one measurements $\boldsymbol{a}^\top \boldsymbol{W} \boldsymbol{b} \in \mathbb{R}$ for $O(rd)$ i.i.d. vectors $\boldsymbol{a}, \boldsymbol{b} \sim \mathcal{N}(0, I_d)$, then one can recover the low-rank matrix $\boldsymbol{W}$ with high probability by solving a convex program (Theorem A.2). See Appendix A for more details.

**Theorem 5.1.** *Assume $\boldsymbol{v}^\star \neq \boldsymbol{0}$ and $\mathrm{rank}(\boldsymbol{W}^\star) \leq r$. Then, with access to* VQ, *there is a randomized, polynomial–time algorithm that with probability at least $1 - e^{-\Omega(m)}$ returns $(\widehat{\boldsymbol{W}}, \widehat{\boldsymbol{v}}) = (\boldsymbol{W}^\star, \boldsymbol{v}^\star)$ using $d + m$ queries, where $m = O(rd)$.*

Recovering $\boldsymbol{v}^\star$ is trivial and can be done in the same way as the vanilla problem by querying $X = [e_i^\top]$ for $i = 1, \ldots, d$.

Hence, the main focus is to recover $\boldsymbol{W}^\star$ using rank-one measurements. Fix $\boldsymbol{a}, \boldsymbol{b} \in \mathbb{R}^d$ and query

$$X = \begin{bmatrix} (\boldsymbol{a}+\boldsymbol{b})^\top \\ \boldsymbol{b}^\top \end{bmatrix}.$$

Let $s_1 = (\boldsymbol{a}+\boldsymbol{b})^\top \boldsymbol{W}^\star \boldsymbol{b}$, $s_2 = \boldsymbol{b}^\top \boldsymbol{W}^\star \boldsymbol{b}$. Then $s_1 - s_2 = \boldsymbol{a}^\top \boldsymbol{W}^\star \boldsymbol{b}$, and the attention weight on the first row equals

$$\alpha = \frac{e^{s_1}}{e^{s_1} + e^{s_2}} = \sigma(s_1 - s_2) = \sigma(\boldsymbol{a}^\top \boldsymbol{W}^\star \boldsymbol{b}).$$

Since $X\boldsymbol{v}^\star = [(\boldsymbol{a}+\boldsymbol{b})^\top \boldsymbol{v}^\star, \ \boldsymbol{b}^\top \boldsymbol{v}^\star]^\top$, the oracle returns $y = \boldsymbol{b}^\top \boldsymbol{v}^\star + \alpha\,(\boldsymbol{a}^\top \boldsymbol{v}^\star)$. If $\boldsymbol{a}^\top \boldsymbol{v}^\star \neq 0$ (this occurs almost surely for Gaussian $\boldsymbol{a}$ when $\boldsymbol{v}^\star \neq 0$), then

$$\alpha = \frac{y - \boldsymbol{b}^\top \boldsymbol{v}^\star}{\boldsymbol{a}^\top \boldsymbol{v}^\star} \in (0, 1),$$

$$t := \sigma^{-1}(\alpha) = \boldsymbol{a}^\top \boldsymbol{W}^\star \boldsymbol{b} = \langle \boldsymbol{a}\boldsymbol{b}^\top, \boldsymbol{W}^\star \rangle.$$

Thus *one* two–row VQ produces a rank–one sensing matrix $\boldsymbol{A} := \boldsymbol{a}\boldsymbol{b}^\top$ and its linear observation $t = \langle \boldsymbol{A}, \boldsymbol{W}^\star \rangle$.

Draw i.i.d. $(\boldsymbol{a}_k, \boldsymbol{b}_k) \sim \mathcal{N}(0, I_d) \times \mathcal{N}(0, I_d)$, and for each pair query the oracle to compute

$$t_k = \sigma^{-1}\left( \frac{y_k - \boldsymbol{b}_k^\top \widehat{\boldsymbol{v}}}{\boldsymbol{a}_k^\top \widehat{\boldsymbol{v}}} \right) = \langle \boldsymbol{a}_k \boldsymbol{b}_k^\top, \boldsymbol{W}^\star \rangle, \quad k = 1, \dots, m.$$

Let $\mathcal{A}$ be the measurement operator with rows $\boldsymbol{A}_k = \boldsymbol{a}_k \boldsymbol{b}_k^\top$. We have gathered the noiseless ROP system $\mathcal{A}(\boldsymbol{W}^\star) = t \in \mathbb{R}^m$. Solve the convex program

$$\widehat{\boldsymbol{W}} \in \arg \min_{\boldsymbol{W} \in \mathbb{R}^{d \times d}} \|\boldsymbol{W}\|_* \qquad (2)$$

$$\text{s.t.} \quad \langle \boldsymbol{a}_k \boldsymbol{b}_k^\top, \boldsymbol{W} \rangle = t_k \ (k = 1, \dots, m).$$

By Theorem A.2 (with $d_1 = d_2 = d$), if $m \geq C\, r(2d)$ then, with probability at least $1 - e^{-\Omega(m)}$ over $\{(\boldsymbol{a}_k, \boldsymbol{b}_k)\}$, the operator $\mathcal{A}$ satisfies RUB with the required condition. Hence, by Theorem A.2, we have $\widehat{\boldsymbol{W}} = \boldsymbol{W}^\star$. This proves Theorem 5.1.

The first part uses $d$ VQs and the recovery of $\boldsymbol{W}^\star$ uses $m$ queries. Thus, the total is $d + m = O(rd)$. Program (2) is convex and solvable in polynomial time.

# 6. Robustness and Stability of Learning

We study a variant of the learning problem where the learner only has access to *approximate* value queries rather than exact ones. The primary motivation is that the algorithm for the noiseless case makes use of $\sigma^{-1}(\cdot)$ function which is not Lipschitz. Hence, if a teacher or an API access provider adds even a tiny amount of noise to the labels or values, then the output of the algorithm can change significantly and the guarantees do not hold. In this section, we show that, even with approximate values, with some assumptions on the norms of the parameters, we can still obtain some reasonable guarantees.

**Approximate value query oracle.** Fix a target function $f^\star : \mathcal{X} \to \mathbb{R}$ (here $f^\star = f_{\boldsymbol{W}^\star, \boldsymbol{v}^\star}$). Given an input $X$ and tolerance $\tau > 0$, the oracle returns $\text{AVQ}(X; \tau)$ satisfying

$$\text{AVQ}(X; \tau) \in [f^\star(X) - \tau, \ f^\star(X) + \tau],$$

and is deterministic (repeated queries on the same $X$ return the same value). The setting also resembles the widely studied statistical query oracle (Kearns, 1998) but is distribution-independent.

**Setup assumptions.** We assume that the norms of the parameters are bounded, $\|\boldsymbol{W}^\star\|_F \leq W$ for a known scalar $W \geq 2$, $\|\boldsymbol{v}^\star\|_2 \leq 1$ (or any constant). Additionally, we assume a margin for the value vector. Let

$\mathcal{I} \subseteq [d]$ such that $|\boldsymbol{v}_i^\star| > 0$ for all $i \in \mathcal{I}$, then we assume $\min_{i \in \mathcal{I}} |\boldsymbol{v}_i^\star| \geq \mu > 0$. Note that sparse vectors are permitted; we only assume that the nonzero entries cannot be arbitrarily close to 0.

**Notation.** Let $\sigma(t) = 1/(1 + e^{-t})$ and $\sigma^{-1}(p) = \log(p/(1-p))$. For $\tau \in (0, 1/2)$ define $\text{clip}(u; \tau, 1 - \tau) = \min\{\max\{u, \tau\}, 1 - \tau\}$.

Let $\tau_{\text{clip}} = \sigma(-\frac{1}{2})$. We will make use of the following elementary helper Lemma (see Lemma A.1), which shows that for all $\widehat{\alpha} \in \mathbb{R}$ and $\alpha^\star \in [\tau_{\text{clip}}, 1 - \tau_{\text{clip}}]$, we have,

$$|\sigma^{-1}(\text{clip}(\widehat{\alpha}; \tau_{\text{clip}}, 1 - \tau_{\text{clip}})) - \sigma^{-1}(\alpha^\star)| \leq 5 |\widehat{\alpha} - \alpha^\star|.$$

The Lemma uses the fact that $\sigma^{-1}(\cdot)$ is Lipschitz within a certain region, even if it is not globally Lipschitz.

**Theorem 6.1.** *Fix $d \geq 1$ and let $f^\star = f_{\boldsymbol{W}^\star, \boldsymbol{v}^\star}$ with $\|\boldsymbol{W}^\star\|_F \leq W$ for a known $W \geq 2$, $\|\boldsymbol{v}^\star\|_2 \leq 1$. Assume $\boldsymbol{v}^\star \neq \boldsymbol{0}$, let $\mathcal{I} = \{i \mid |\boldsymbol{v}_i^\star| > 0\}$ and $\min_{i \in \mathcal{I}} |\boldsymbol{v}_i^\star| \geq \mu > 0$. Assume that the learner has access to a deterministic oracle $\text{AVQ}(\cdot; \tau)$ returning a value within $\pm \tau$ of $f^\star(\cdot)$. Then for any $\epsilon_v, \epsilon_W \in (0, 1)$ there is a polynomial-time algorithm that makes $O(d^2)$ approximate value queries, each with tolerance $\tau = O\left( \min\left\{ \mu, \frac{\epsilon_v}{\sqrt{d}}, \frac{\mu\, \epsilon_W}{W^2 d} \right\} \right)$ and outputs $(\widehat{\boldsymbol{v}}, \widehat{\boldsymbol{W}})$ such that*

$$\|\widehat{\boldsymbol{v}} - \boldsymbol{v}^\star\|_2 \leq \epsilon_v \qquad and \qquad \|\widehat{\boldsymbol{W}} - \boldsymbol{W}^\star\|_F \leq \epsilon_W.$$

Approximating $\boldsymbol{v}^\star$ is straightforward; the key challenge is to approximate $\boldsymbol{W}^\star$. The algorithm is almost exactly the same as the noiseless case, but the probes are scaled. The key technical idea is to construct probes such that $\sigma^{-1}(\cdot)$ is always applied to values within certain bounds where it is Lipschitz, and consequently, the error in the remaining operations can be controlled to achieve the desired guarantee.

*Proof.* To approximate $\boldsymbol{v}^\star$, we can simply query basis vectors, $X = [\boldsymbol{e}_i^\top]$ for each $i \in [d]$ with tolerance $\tau_v$ and set

$$\widehat{\boldsymbol{v}}_i := \text{AVQ}([\boldsymbol{e}_i^\top]; \tau_v).$$

Then $|\widehat{\boldsymbol{v}}_i - \boldsymbol{v}_i^\star| \leq \tau_v$ for all $i$, and therefore $\|\widehat{\boldsymbol{v}} - \boldsymbol{v}^\star\|_2 \leq \sqrt{d}\, \tau_v$ and thus less than $\epsilon_v$ for $\tau_v \leq \epsilon_v / \sqrt{d}$. If additionally $\tau_v \leq \mu/2$, then $|\widehat{\boldsymbol{v}}_i| \geq \mu/2$ for every $i \in \mathcal{I}$, which will be used to control ratios below. If $|\widehat{\boldsymbol{v}}_i| \leq \mu/2$, then we know that $\boldsymbol{v}_i^\star = 0$.

*Approximating $\boldsymbol{W}^\star$.* We want to estimate $\boldsymbol{W}^\star$ such that $\|\widehat{\boldsymbol{W}} - \boldsymbol{W}^\star\|_F \leq \epsilon_W$. Fix $a = \frac{1}{2}$ and $b = 1/W$ so that $|ab\, \boldsymbol{W}_{ij}^\star| \leq ab \|\boldsymbol{W}^\star\|_F \leq \frac{1}{2}$. For a two-row input $X = \begin{bmatrix} u^\top \\ x^\top \end{bmatrix}$, we can write

$$f_{\boldsymbol{W}, \boldsymbol{v}}(X) = x^\top \boldsymbol{v} + \alpha_1 (u^\top \boldsymbol{v} - x^\top \boldsymbol{v}), \qquad (3)$$

$$\alpha_1 = \sigma\big(u^\top W x - x^\top W x\big). \qquad (4)$$

Let $p \in [d]$ be any entry such that $|v_p^\star| > \mu$. To estimate entry $(i,j)$ of $W^\star$, use the probe

$$X = \begin{bmatrix} (b\boldsymbol{u} + a\mathbf{e}_j)^\top \\ (a\mathbf{e}_j)^\top \end{bmatrix}$$

where $\boldsymbol{u} = \mathbf{e}_i$ if $|v_i^\star| \geq \mu$ and $\boldsymbol{u} = \mathbf{e}_i + \mathbf{e}_p$ if $v_i^\star = 0$. Let's first consider the case where $|v_i^\star| \geq \mu$. Plugging it to (3) we have,

$$f^\star(X) = a\,\boldsymbol{v}_j^\star + \alpha_1^\star\,(b\,\boldsymbol{v}_i^\star),$$

$$\alpha_1^\star := \alpha_1(X; W^\star) = \frac{f^\star(X) - a\,\boldsymbol{v}_j^\star}{b\,\boldsymbol{v}_i^\star}.$$

On the other hand, the logit term in (4) simplifies to $(b\mathbf{e}_i + a\mathbf{e}_j)^\top W^\star(a\mathbf{e}_j) - (a\mathbf{e}_j)^\top W^\star(a\mathbf{e}_j) = ab\,W_{ij}^\star$, so

$$\alpha_1^\star = \sigma(ab\,W_{ij}^\star) \qquad \Longrightarrow \qquad W_{ij}^\star = \frac{1}{ab}\,\sigma^{-1}(\alpha_1^\star).$$

Since $ab\,W_{ij}^\star \in [-\frac{1}{2}, \frac{1}{2}]$, we have $\alpha_1^\star \in [\sigma(-\frac{1}{2}), 1 - \sigma(-\frac{1}{2})]$. Let $\tau_{\text{clip}} := \sigma(-\frac{1}{2})$ and $L_{\tau_{\text{clip}}} := \frac{1}{\tau_{\text{clip}}(1-\tau_{\text{clip}})} \leq 5$.

**Estimators from approximate values.** Fix $(i,j)$ and let $X$ be the probe above. Query the oracle once on $X$ with tolerance $\tau_f$ and define

$$\widehat{f}(X) := \text{AVQ}(X; \tau_f) = f^\star(X) + \eta_f, \qquad |\eta_f| \leq \tau_f.$$

Use the coordinate estimates $\widehat{\boldsymbol{v}}_i, \widehat{\boldsymbol{v}}_j$ obtained earlier with tolerance $\tau_v$, so $\widehat{\boldsymbol{v}}_k = \boldsymbol{v}_k^\star + \eta_k$ with $|\eta_k| \leq \tau_v$. Define

$$\widehat{\alpha}_1 := \frac{\widehat{f}(X) - a\,\widehat{\boldsymbol{v}}_j}{b\,\widehat{\boldsymbol{v}}_i}. \qquad (5)$$

We have

$$\widehat{\alpha}_1 - \alpha_1^\star = \frac{\widehat{f}(X) - a\,\widehat{\boldsymbol{v}}_j}{b\,\widehat{\boldsymbol{v}}_i} - \alpha_1^\star$$

$$= \frac{\widehat{f}(X) - a\,\boldsymbol{v}_j^\star + a(\boldsymbol{v}_j^\star - \widehat{\boldsymbol{v}}_j) - \alpha_1^\star b\,\widehat{\boldsymbol{v}}_i}{b\,\widehat{\boldsymbol{v}}_i},$$

and adding and subtracting $\alpha_1^\star b\,\boldsymbol{v}_i^\star$ gives $\widehat{\alpha}_1 - \alpha_1^\star$

$$= \frac{\widehat{f}(X) - a\,\boldsymbol{v}_j^\star - \alpha_1^\star b\,\boldsymbol{v}_i^\star + a(\boldsymbol{v}_j^\star - \widehat{\boldsymbol{v}}_j) + \alpha_1^\star b(\boldsymbol{v}_i^\star - \widehat{\boldsymbol{v}}_i)}{b\,\widehat{\boldsymbol{v}}_i}.$$

Since $f^\star(X) = a\,\boldsymbol{v}_j^\star + \alpha_1^\star b\,\boldsymbol{v}_i^\star$, the first three terms equal $\eta_f$, hence

$$|\widehat{\alpha}_1 - \alpha_1^\star| \leq \left| \frac{\eta_f + a(\boldsymbol{v}_j^\star - \widehat{\boldsymbol{v}}_j) + \alpha_1^\star b(\boldsymbol{v}_i^\star - \widehat{\boldsymbol{v}}_i)}{b\,\widehat{\boldsymbol{v}}_i} \right| \qquad (6)$$

$$\leq \frac{|\eta_f| + a\,|\widehat{\boldsymbol{v}}_j - \boldsymbol{v}_j^\star|}{|b\,\widehat{\boldsymbol{v}}_i|} + \frac{|\widehat{\boldsymbol{v}}_i - \boldsymbol{v}_i^\star|}{|\widehat{\boldsymbol{v}}_i|}.$$

**From tolerances to accuracy.** For $\tau_v \leq \mu/2$, we have $|\widehat{\boldsymbol{v}}_i| \geq \mu/2$ for all $i$. Then

$$\frac{1}{|b\,\widehat{\boldsymbol{v}}_i|} \leq \frac{2}{b\mu} = \frac{2W}{\mu}, \qquad \frac{|\widehat{\boldsymbol{v}}_i - \boldsymbol{v}_i^\star|}{|\widehat{\boldsymbol{v}}_i|} \leq \frac{2}{\mu}\,\tau_v.$$

Using $|\eta_f| \leq \tau_f$ and $|\widehat{\boldsymbol{v}}_j - \boldsymbol{v}_j^\star| \leq \tau_v$ in (6) yields

$$|\widehat{\alpha}_1 - \alpha_1^\star| \leq \frac{2W}{\mu}\big(\tau_f + a\,\tau_v\big) + \frac{2}{\mu}\tau_v = \frac{2W}{\mu}\tau_f + \frac{W+2}{\mu}\tau_v.$$

With the choices

$$\tau_f \leq \frac{\mu\,\epsilon_W}{80\,W^2 d}, \qquad \tau_v \leq \min\left\{\frac{\mu}{2}, \frac{\mu\,\epsilon_W}{80\,W^2 d}\right\},$$

we obtain $|\widehat{\alpha}_1 - \alpha_1^\star| \leq \epsilon_W/(10Wd)$. Define

$$\widehat{W}_{ij} := \frac{1}{ab}\,\sigma^{-1}\big(\text{clip}(\widehat{\alpha}_1; \tau_{\text{clip}}, 1 - \tau_{\text{clip}})\big).$$

By Lemma A.1,

$$|\widehat{W}_{ij} - W_{ij}^\star| \leq \frac{L_{\tau_{\text{clip}}}}{ab}\,|\widehat{\alpha}_1 - \alpha_1^\star| \leq \frac{\epsilon_W}{d} \quad \text{for all } (i,j),$$

and hence $\|\widehat{W} - W^\star\|_F \leq \epsilon_W$.

The analysis for the case when $\boldsymbol{v}_i^\star = 0$ is quite similar. With probe vector $\boldsymbol{u} = \mathbf{e}_i + \mathbf{e}_p$, we have that,

$$W_{ij}^\star = \big(\frac{1}{ab}\sigma^{-1}(\alpha^\star) - W_{pj}^\star\big)$$

and similarly, we will compute $\widehat{W}_{ij} = \big(\frac{1}{ab}\sigma^{-1}(\widehat{\alpha}) - \widehat{W}_{pj}\big)$. The attention weight at the first position (5) would instead be

$$\widehat{\alpha}_1 := \frac{\widehat{f}(X) - a\,\widehat{\boldsymbol{v}}_j}{b\,\widehat{\boldsymbol{v}}_p}$$

where $|\widehat{\boldsymbol{v}}_p| \geq \mu/2$ and we can obtain $\frac{L_{\tau_{\text{clip}}}}{ab}\,|\widehat{\alpha}_1 - \alpha_1^\star| \leq \frac{\epsilon_W}{d}$ by following the same line of analysis and tolerance values. We then have $|\widehat{W}_{ij} - W_{ij}^\star|$

$$= \left| \frac{1}{ab}\,\sigma^{-1}\big(\text{clip}_{\tau_{\text{clip}}}(\widehat{\alpha}_1)\big) - \widehat{W}_{pj} - \big(\frac{1}{ab}\sigma^{-1}(\alpha_1^\star) - W_{pj}^\star\big) \right|$$

$$\leq \frac{L_{\tau_{\text{clip}}}}{ab}\,|\widehat{\alpha}_1 - \alpha_1^\star| + |W_{pj}^\star - \widehat{W}_{pj}| \leq \frac{2\epsilon_W}{d}.$$

Dividing the tolerance values for the first case by 2 gives the desired accuracy.

The reconstruction uses $d$ one-row queries and $d^2$ two-row probe queries, for a total of $d + d^2$ queries. The only requirement is that the oracle can answer these queries with an additive error at most $\tau_v, \tau_f = O\big(\frac{\mu\,\epsilon_W}{W^2 d}\big)$ (up to constants), and $\tau_v \leq \mu/2$ to ensure the denominator in (6) is bounded away from 0. $\qquad \square$

**Membership queries.** So far, our analysis has focused on value queries where the oracle returns real-valued scalars. In App. B we further consider a strictly weaker oracle in which each query returns only a binary response $\mathrm{MQ}(X) \in \{0, 1\}$ indicating whether $f^\star(X)$ is positive. The main challenge is that the key step used for value queries, namely recovering an attention weight and inverting the sigmoid, is no longer available; instead, the learner only observes the sign of $f^\star(X)$.

Despite this loss of information, we show that the parameters remain efficiently recoverable, though under somewhat stronger assumptions on the vector $\boldsymbol{v}^\star$. At a high level, to estimate each entry $\boldsymbol{W}_{ij}^\star$ we use the same two-row probe family as in Theorem 6.1, but view it as a one-parameter set of queries indexed by a scalar $a > 0$ with $ab$ fixed so that the logit remains bounded. Under this parametrisation, the membership response flips sign exactly once as a function of $a$, and a bisection search locates the corresponding threshold; this threshold estimate can then be algebraically converted into an additive estimate of $\boldsymbol{W}_{ij}^\star$. See Appendix B for details.

# 7. Learnability of Multi-head Attention with Queries

In this section, we consider the problem of learning a multi-head attention layer. Let $d_h$ denote the embedding dimension of each head, and $d$ the total embedding dimension, such that the number of heads is $H = d/d_h$. The parameter matrices for the multi-head attention are given by $\boldsymbol{W}^{(1)}, \ldots, \boldsymbol{W}^{(H)} \in \mathbb{R}^{d \times d}$, and the corresponding projection vectors by $\boldsymbol{v}^{(1)}, \ldots, \boldsymbol{v}^{(H)} \in \mathbb{R}^d$. Let $W$ and $v$ denote the sets of all weight matrices and vectors, respectively. The output of the multi-head attention layer is then defined as (see App. C for details of the formulation)

$$f_{W,v}^H(X) = \sum_{h=1}^H \boldsymbol{\alpha}^{(h)}(X, \boldsymbol{W}^{(h)})^\top (X\boldsymbol{v}^{(h)}).$$

where $\boldsymbol{\alpha}^{(h)}(X, \boldsymbol{W}^{(h)})$ represents the attention weights computed for head $h$.

We observe that, unlike the single-head attention layer, not every set of parameters $(W, v)$ corresponds to a unique function $f_{W,v}^H(X)$. In particular, we show the following proposition.

**Proposition 7.1.** *Let $f_{W,v}^H$ be a multi-head attention layer with $H$ heads and with a set of parameters $(W, v)$. Then, there exist multiple sets of parameters $(W, v)$ that lead to the same function $f_{W,v}^H : \mathbb{R}^{N \times d} \to \mathbb{R}$.*

*Proof.* We consider the set of weight parameters $W$ such that the weight matrices of all heads are identical, i.e.,

$\boldsymbol{W}^{(h)} = \boldsymbol{A}$ for all $h \in [H]$, where $\boldsymbol{A}$ is any fixed matrix. Let $\boldsymbol{b} \in \mathbb{R}^d$ be any arbitrary non zero vector and let $\lambda_1, \ldots, \lambda_H$ be $H$ non-negative weights such that $\sum_{i=1}^H \lambda_i = 1$. For any such weights, set $\boldsymbol{v}^{(h)} = \lambda_h \boldsymbol{b}$ for all $h \in [H]$. Then, the multi-head attention model computes,

$$f_{W,v}^H(X) = \sum_{h=1}^H \boldsymbol{\alpha}^{(h)}(X, \boldsymbol{W}^{(h)})^\top (X\boldsymbol{v}^{(h)})$$

$$= \boldsymbol{\alpha}(X, \boldsymbol{A})^\top \sum_{h=1}^H \lambda_h (X\boldsymbol{b}) = \boldsymbol{\alpha}(X, \boldsymbol{A})^\top (X\boldsymbol{b}).$$

Thus, for different weights $\lambda_1, \ldots, \lambda_H$, we will have different projection vectors $\boldsymbol{v}^{(1)}, \ldots, \boldsymbol{v}^{(H)}$ but all of them would lead to the same function as defined above. $\square$

## 7.1. Discussion

As a consequence of Proposition 7.1, the parameters of the multi-head attention layer cannot be uniquely identified from value queries alone, and achieving guarantees like Theorem 4.1 is not possible. For exact learning with queries to be feasible, the parameter-function map needs to be one-to-one. Different from efficient learning, we say a class of functions is identifiable from queries if there exists some set of input-output pairs that uniquely determines the target function.

Learnability of multi-head attention with queries is left for future work, and many questions around that remain open. We discuss some potential directions and approaches to tackle the multi-head problem. (i) It could be useful to first identify structural assumptions under which the class is identifiable. As a first step, in Appendix C, we show that if the parameters of each head $(\boldsymbol{W}^{(h)}, \boldsymbol{v}^{(h)})$ lie in mutually orthogonal subspaces, then the class is identifiable (up to permutation). We also conjecture that under the orthogonal subspace assumption, multi-head attention could be learnable with queries by leveraging the structure of the Hessian with respect to probe vectors (see App. C for more details). (ii) Apart from parameter estimation, an interesting direction is to understand functional equivalence. Given access to queries, is it possible to learn a function that closely approximates the target even if the parameters are different? Even basic decidability questions around this are not clearly understood — for instance, given two sets of parameters of multi-head attention models, is the problem of checking whether they represent the same function decidable? (iii) Further, it could be interesting to analyse whether distribution-dependent PAC guarantees (such as Chen et al. (2021) for MLPs) are attainable, unlike distribution-independent ones for single-head attention.

## Impact Statement

This work takes a step toward understanding how the weights of attention-based and Transformer models can be recovered using only black-box access to their outputs. While the analysis focuses on a somewhat simplified setting and does not have immediate implications, it offers insights into the broader problem of weight extraction (or model stealing), which has important implications for AI security. Such insights can help identify potential API vulnerabilities and guide efforts to prevent model stealing. Beyond that, it also helps understand which components of Transformers are identifiable solely from input-output mappings.

## Acknowledgments

We would like to thank Charles London and the anonymous reviewers for their thoughtful feedback on this work. MH gratefully acknowledges the stimulating research environment of the GRK 2853/1 "Neuroexplicit Models of Language, Vision, and Action", funded by the Deutsche Forschungsgemeinschaft (DFG, German Research Foundation) under project number 471607914.

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

## A. Technical Tools

**Lemma A.1.** *If $\alpha^\star \in [\tau_{\mathrm{clip}}, 1 - \tau_{\mathrm{clip}}]$ with $\tau_{\mathrm{clip}} = \sigma(-\frac{1}{2})$, then for all $\widehat{\alpha} \in \mathbb{R}$,*

$$\left| \sigma^{-1}(\mathrm{clip}(\widehat{\alpha}; \tau_{\mathrm{clip}}, 1 - \tau_{\mathrm{clip}})) - \sigma^{-1}(\alpha^\star) \right| \leq L_{\tau_{\mathrm{clip}}} |\widehat{\alpha} - \alpha^\star| \leq 5 |\widehat{\alpha} - \alpha^\star|.$$

*Proof.* On $(0,1)$, $(\sigma^{-1})'(p) = \frac{1}{p(1-p)}$, hence on $[\tau_{\mathrm{clip}}, 1 - \tau_{\mathrm{clip}}]$ the derivative is at most $L_{\tau_{\mathrm{clip}}} = \frac{1}{\tau_{\mathrm{clip}}(1 - \tau_{\mathrm{clip}})}$. Also, $u \mapsto \mathrm{clip}(u; \tau_{\mathrm{clip}}, 1 - \tau_{\mathrm{clip}})$ is 1-Lipschitz and maps into $[\tau_{\mathrm{clip}}, 1 - \tau_{\mathrm{clip}}]$. Applying the mean value theorem to $\sigma^{-1}$ on this interval leads to the claim, and $L_{\tau_{\mathrm{clip}}} \leq 5$ gives the desired bound. □

### A.1. Background: Matrix Sensing

For matrices $A, B$, we write $\langle A, B \rangle = \mathrm{tr}(A^\top B)$ for the Frobenius inner product on $\mathbb{R}^{d \times d}$, and $\|W\|_*$ for the nuclear norm (sum of singular values). For vectors $a, b \in \mathbb{R}^d$, we use $a \otimes b := ab^\top$ to denote a rank–one matrix.

**Matrix sensing and the ROP model.** A matrix sensing problem specifies an unknown matrix $W^\star \in \mathbb{R}^{d_1 \times d_2}$ and a linear map

$$\mathcal{A} : \mathbb{R}^{d_1 \times d_2} \to \mathbb{R}^m, \qquad (\mathcal{A}(W))_k = \langle A_k, W \rangle \quad (1 \leq k \leq m),$$

with known sensing matrices $A_k$. Cai & Zhang (2015) introduced the "rank-one projection" (ROP) model that takes $A_k = a_k b_k^\top$ with $a_k \in \mathbb{R}^{d_1}$, $b_k \in \mathbb{R}^{d_2}$ (typically i.i.d. Gaussian), so each measurement is

$$y_k = \langle a_k b_k^\top, W^\star \rangle = a_k^\top W^\star b_k.$$

They also introduce a concept of Restricted uniform boundedness (RUB) property (Cai & Zhang, 2015, Definition 2.1), which is analogous to the Restricted Isometry Property (RIP) but more suited to the ROP model. They show that the RUB property leads to guaranteed recovery of low-rank matrices. In particular, given a collection of matrices $\mathcal{A}$ satisfying the RUB property and measurements $y = \mathcal{A}(W^\star)$, one can recover a low-rank matrix $W$ by solving a convex program which minimizes the nuclear norm $\min_{W \in \mathbb{R}^{d_1 \times d_2}} \|W\|_*$ under the constraints that $\langle A_k, W \rangle = y_k$ for all $k$.

The final useful fact is that constructing rank-one projections by randomly sampling i.i.d. from the standard Gaussian distribution $\mathcal{N}(0, I_d)$ leads to a collection $\mathcal{A} = \{A_1, \ldots, A_m\}$ that satisfies the RUB property with probability at least $1 - e^{-\Omega(m)}$. See Section 2.1 in Cai & Zhang (2015) for more details on the background described in this section.

**Theorem A.2** (Recovery under ROP; Cai & Zhang, 2015, Thm. 2.2 and Cor. 2.1). *Let $A_k = a_k b_k^\top$ with $a_k \in \mathbb{R}^{d_1}$, $b_k \in \mathbb{R}^{d_2}$ i.i.d. standard Gaussian for $k = 1, \ldots, m$ where $m = \Omega(r(d_1 + d_2))$. For some unknown $W \in \mathbb{R}^{d_1 \times d_2}$, the measurements are of the form $(\mathcal{A}(W)) = \langle A_k, W \rangle$. Then with probability at least $1 - e^{-\Omega(m)}$, the operator $\mathcal{A}$ satisfies RUB and the following convex program recovers any rank $\leq r$ matrix $W$ exactly,*

$$\min_{W \in \mathbb{R}^{d_1 \times d_2}} \|W\|_* \quad s.t. \quad \langle A_k, W \rangle = y_k \ (k = 1, \ldots, m). \tag{7}$$

## B. Learning Attention Head from Membership Queries

We want to identify the target parameters $(W^\star, v^\star)$ when the learner has access to a membership query oracle $\mathrm{MQ}_{W*,v*} : \mathbb{R}^{N \times d} \to \{0, 1\}$ where

$$\mathrm{MQ}_{W*,v*}(X) := \begin{cases} 1 & \text{if } f_{W*,v*}(X) > 0 \\ 0 & \text{else} \end{cases} \tag{8}$$

where, as above

$$f_{W,v}(X) = \alpha(X, W)^\top (X v) \tag{9}$$

Note that we can only expect to identify the direction of $v$, but not its magnitude. We may thus assume, without loss of generality, that $\|v^\star\| = 1$.

We will work with

$$h_{W*,v*}(X) := \mathrm{sign}(f_{W*,v*}(X)) \tag{10}$$

and use

$$h_{W*,v*}(X) = 2\,\mathrm{MQ}_{W*,v*}(X) - 1 \tag{11}$$

**Assumptions.** We assume: $\|W^\star\|_F \leq W$ for a known scalar $W \geq 2$, $\|v^\star\|_2 = 1$, and a value margin $\min_{i \in [d]} |v_i^\star| \geq \mu > 0$. Note that the value margin assumption rules out sparse vectors and is stronger than the assumptions of Theorem 6.1, but we hypothesise that it should be possible to extend it to sparse vectors as well.

**Theorem B.1** (Recovery from membership queries). *Fix $d \geq 1$ and let $f^\star = f_{W^\star,v^\star}$ with $\|W^\star\|_F \leq W$ for a known $W \geq 2$, $\|v^\star\|_2 = 1$, and $\min_{i \in [d]} |v_i^\star| \geq \mu > 0$. Assume that the learner has access to a deterministic oracle $\mathrm{MQ}(\cdot)$ returning 1 if $f^\star(\cdot) > 0$, and 0 otherwise. Then for any $\delta > 0$ and for any $\epsilon_W \in (0,1)$ there is a polynomial-time algorithm that makes*

$$\mathcal{O}\left( \frac{d}{\epsilon_v^2} \log \frac{d^2}{\delta} + d^2 \log \left( C \cdot \frac{W^{3/2} d}{\epsilon_W \mu^{7/2}} \right) \right)$$

*randomized membership queries (with $C$ an absolute constant, independent of $W$) where $\epsilon_v := \frac{\epsilon_W \mu^3}{40 W d}$, with outputs $(\widehat{v}, \widehat{W})$ such that*

$$\|\widehat{v} - v^\star\|_\infty \leq \epsilon_v \qquad \text{and} \qquad \|\widehat{W} - W^\star\|_F \leq \epsilon_W.$$

*with probability $1 - \delta$ over the randomized queries.*

**Approximating $v^\star$.** With length-one queries, the problem is essentially equivalent to learning a halfspace with membership queries which is well-studied. In this setting, we can use random queries with $x$ from the unit sphere, and use the identity (for any nonzero $v$):

$$\mathbb{E}_{x \sim \mathrm{Uniform}(\mathbb{S}^{d-1})}[\mathrm{sign}(v^\top x)x] = R \cdot \frac{v}{\|v\|} \tag{12}$$

where $R := \frac{\Gamma(d/2)}{\sqrt{\pi}\Gamma((d+1)/2)} = \Theta(1/\sqrt{d})$. Thus

$$\mathbb{E}_{x \sim \mathrm{Uniform}(\mathbb{S}^{d-1})}[(2\,\mathrm{MQ}_{W^\star,v^\star}(x) - 1)x] = \mathbb{E}_{x \sim \mathrm{Uniform}(\mathbb{S}^{d-1})}[h_{W^\star,v^\star}(x)x] = \mathbb{E}_{x \sim \mathrm{Uniform}(\mathbb{S}^{d-1})}[\mathrm{sign}(v^\top x)x] \tag{13}$$

where the first equality holds provided the probability of $f_{W^\star,v^\star}(x) = 0$ is zero; this is true provided $v \neq 0$, which is true by assumption.

Let $k$ be the number of queries, and let $x_1, \ldots, x_k \in \mathbb{R}^d$ be i.i.d. query vectors sampled as $x_i \sim \mathrm{Uniform}(\mathbb{S}^{d-1})$, and estimate

$$\widehat{v} := \frac{1}{R}\frac{1}{k}\sum_{i=1}^k (2\,\mathrm{MQ}_{W^\star,v^\star}(x_i) - 1)x_i =^{(*)} \frac{1}{R}\frac{1}{k}\sum_{i=1}^k \mathrm{sign}((v^\star)^\top x_i)x_i \tag{14}$$

where the equality $(*)$ holds a.s. for the choice of the query vectors.

By Hoeffding's inequality, for any coordinate $j = 1, \ldots d$,

$$\mathbb{P}\left( |\widehat{v}_j - v_j^\star| \geq t \right) \leq 2\exp\left( -\frac{2t^2 R^2 k}{4} \right) \tag{15}$$

Thus $\|\widehat{v} - v^\star\|_\infty \leq t$ with probability $1 - d \cdot 2\exp\left(-\frac{t^2 R^2 k}{2}\right)$. Substitute $\delta := 2d\exp\left(-\frac{t^2 R^2 k}{2}\right)$, hence $\frac{2}{t^2 R^2}\log\frac{2d}{\delta} = k$. Thus, with $k = O\left(\frac{1}{R^2 \epsilon_v^2}\log\frac{d^2}{\delta}\right) = O\left(\frac{d}{\epsilon_v^2}\log\frac{d^2}{\delta}\right)$ queries, we get $L_\infty$ error $\leq \epsilon_v$ with probability $1 - \delta$. Note that, a query $\mathrm{sign}(v^\top x)$ is scale-invariant and one cannot hope to recover $\|v\|_2$ regardless of the number of queries.

**Approximating $W^\star$.** Define $\eta_W = \frac{\epsilon_W}{d}$.

We want to estimate $W^\star$ such that $\|\widehat{W} - W^\star\|_F \leq d\eta_W = \epsilon_W$. As in the noise robustness case, for a two-row input $X = \begin{bmatrix} u^\top \\ x^\top \end{bmatrix}$, we have,

$$f_{W,v}(X) = x^\top v + \alpha_1(u^\top v - x^\top v), \qquad \alpha_1 = \sigma(u^\top W x - x^\top W x).$$

To estimate the entry $(i, j)$ of $\boldsymbol{W}^\star$, we will use the following input as a probe,

$$\boldsymbol{X} = \begin{bmatrix} (b\mathbf{e}_i + a\mathbf{e}_j)^\top \\ (a\mathbf{e}_j)^\top \end{bmatrix},$$

where $\mathbf{e}_i$ is the ith standard basis vector or one-hot vector. Then, we have

$$f_{\boldsymbol{W}^\star, \boldsymbol{v}^\star}(\boldsymbol{X}) = a\,\boldsymbol{v}_j^\star + \alpha_1^\star \cdot b\,\boldsymbol{v}_i^\star$$

Unlike in the Noise Robustness case, here we just observe a thresholded value

$$\mathrm{MQ}_{\boldsymbol{W}^\star, \boldsymbol{v}^\star}(X) = H(f_{\boldsymbol{W}^\star, \boldsymbol{v}^\star}(X)) = H(a\,\boldsymbol{v}_j^\star + \alpha_1^\star \cdot b \cdot \boldsymbol{v}_i^\star), \qquad \alpha_1^\star = \sigma(ab\boldsymbol{W}_{ij}^\star),$$

where $H(x) := \mathbb{1}_{x>0}$.

We want to identify $\boldsymbol{W}_{ij}^\star$ by locating the sign change, i.e., finding $a, b$ such that

$$0 = a\,\boldsymbol{v}_j^\star + \alpha_1^\star \cdot b \cdot \boldsymbol{v}_i^\star \tag{16}$$

and thus

$$\frac{-a\,\boldsymbol{v}_j^\star}{\boldsymbol{v}_i^\star} = \alpha_1^\star \cdot b = \sigma(ab\boldsymbol{W}_{ij}^\star) \cdot b \tag{17}$$

Once we have (approximately) identified such $a, b$, we will be able to (approximately) determine $\boldsymbol{W}_{ij}^\star$ from this equation. Set $\alpha = \frac{\boldsymbol{v}_j^\star}{\boldsymbol{v}_i^\star}$. Note that $a, b$ are under our control in designing the query; $\alpha$ is already known up to a small approximation error; $\boldsymbol{W}_{ij}^\star$ is unknown.

We reparameterize by introducing a constant $s := -\operatorname{sign}(\alpha) \cdot \frac{1}{2W}$, and define $b := s/a$. The problem (17) thus takes the form

$$-a\alpha = \sigma(s\boldsymbol{W}_{ij}^\star)\frac{s}{a} \tag{18}$$

We fix $ab = s$ so the logit $s\boldsymbol{W}_{ij}^\star$ stays bounded. Our queries are from a one-parameter family ($a > 0$):

$$\boldsymbol{X}(a) = \begin{bmatrix} (b\mathbf{e}_i + a\mathbf{e}_j)^\top \\ (a\mathbf{e}_j)^\top \end{bmatrix} = \begin{bmatrix} (\frac{s}{a}\mathbf{e}_i + a\mathbf{e}_j)^\top \\ (a\mathbf{e}_j)^\top \end{bmatrix};$$

The goal is to bisect over $a$ in some interval to approximately identify an $a$ at which $f_{\boldsymbol{W}^\star, \boldsymbol{v}^\star}(\boldsymbol{X}(a)) = 0$. For this, we would like to determine an interval inside which there is guaranteed to be exactly one such solution $a$. Defining

$$F(a) := a\alpha + \frac{\sigma(\boldsymbol{W}_{ij}^\star s)s}{a} \tag{19}$$

we have

$$h_{\boldsymbol{W}^\star, \boldsymbol{v}^\star}(\boldsymbol{X}(a)) = \operatorname{sign}\left(a\boldsymbol{v}_j^\star + \alpha_1^\star b\boldsymbol{v}_i^\star\right) = \operatorname{sign}\left(a\boldsymbol{v}_j^\star + \alpha_1^\star \frac{s}{a}\boldsymbol{v}_i^\star\right) = \operatorname{sign}\left(a\boldsymbol{v}_j^\star + \frac{\sigma(\boldsymbol{W}_{ij}^\star s)s}{a}\boldsymbol{v}_i^\star\right)$$

$$= \operatorname{sign}(\boldsymbol{v}_i^\star)\operatorname{sign}\left(a\frac{\boldsymbol{v}_j^\star}{\boldsymbol{v}_i^\star} + \frac{\sigma(\boldsymbol{W}_{ij}^\star s)s}{a}\right)$$

$$= \operatorname{sign}(\boldsymbol{v}_i^\star)\operatorname{sign}\left(\underbrace{a\alpha + \frac{\sigma(\boldsymbol{W}_{ij}^\star s)s}{a}}_{F(a)}\right)$$

The function $F(a)$ is strictly monotone in $a \in (0, \infty)$, because its derivative is

$$F'(a) = \alpha - \frac{\sigma(\boldsymbol{W}_{ij}^\star s)s}{a^2} \tag{20}$$

and note our construction ensures $\text{sign}(\alpha)\,\text{sign}(s) = -1$, giving this constant sign. Note here that the two terms in $F'(a)$ have the same sign because $\text{sign}(\alpha)\,\text{sign}(s) = -1$. Also, $\lim_{a\downarrow 0} F(a) = \text{sign}(s)\cdot\infty$, $\lim_{a\to\infty} F(a) = \text{sign}(\alpha)\cdot\infty = -\text{sign}(s)\cdot\infty$. Thus, for $a \in (0,\infty)$, $f_{\boldsymbol{W}^\star,\boldsymbol{v}^\star}(\boldsymbol{X}(a))$ has exactly one zero $a^\star$, and $h_{\boldsymbol{W}^\star,\boldsymbol{v}^\star}(\boldsymbol{X}(a))$ flips sign exactly once.

We now want to identify a specific interval of length $0 \leq \tau \leq \sqrt{\frac{1}{2W\mu}}$ (to be fixed later) inside which the sign flip occurs. First, we note that a positive solution $a$ of (18) satisfies:

$$a = \sqrt{-\frac{\sigma(\boldsymbol{W}_{ij}^\star s)s}{\alpha}} = \sqrt{\frac{\sigma(\boldsymbol{W}_{ij}^\star s)}{2W|\alpha|}} \tag{21}$$

which gives us an a-priori upper and lower bound on $a^\star$:

$$a_{max} := \sqrt{\frac{1}{2W\mu}} \geq^{(1)} a^\star = \sqrt{\frac{\sigma(\boldsymbol{W}_{ij}^\star s)}{2W|\alpha|}} \geq^{(2)} \sqrt{\frac{\sigma(-W\frac{1}{2W})\mu}{2W}} \geq^{(3)} \sqrt{\frac{\sigma(-\frac{1}{2})\mu}{2W}} \geq^{(4)} \sqrt{\frac{\mu}{8W}} =: a_{min} \tag{22}$$

where we used $|\alpha| \in [\mu, \frac{1}{\mu}]$ in (1) and (2), $|\boldsymbol{W}_{ij}^\star| \leq \|\boldsymbol{W}^\star\|_F \leq W$, which implies $\boldsymbol{W}_{ij}^\star s \geq -1/2$ because $s = -\text{sign}(\alpha)/(2W)$ in (2), $\sigma(-1/2) \geq 1/4$ in (4).

We are guaranteed that $\text{sign}(F(a_{max}))\,\text{sign}(F(a_{min})) \neq 1$. We can thus bisect within the $[a_{min}, a_{max}]$ until we have arrive at a small interval $[a_-, a_+]$ with $a_+ - a_- \leq \tau \leq \sqrt{\frac{1}{2W\mu}}$ such that $\text{sign}(F(a_-))\,\text{sign}(F(a_+)) \neq 1$; this takes

$$\mathcal{O}(\log \frac{a_{max} - a_{min}}{\tau}) = \mathcal{O}(\log \frac{a_{max}}{\tau}) = \mathcal{O}\left(\log \frac{1}{\sqrt{2W\mu}\cdot\tau}\right) \tag{23}$$

steps, and we are guaranteed that $a_- \leq a^\star \leq a_+$. That is, we find $\widehat{a}$ that has deviation at most $\tau$ from the unique point $a^\star$ where $f_{\boldsymbol{W}^\star,\boldsymbol{v}^\star}(\boldsymbol{X}(a^\star)) = 0$:

$$a^\star = \sqrt{-\frac{\sigma(\boldsymbol{W}_{ij}^\star s)s}{\alpha}} \tag{24}$$

We now explain how to estimate $\boldsymbol{W}_{i,j}^\star$ on the basis of $a^\star$.

Note that the input to $\sigma(\cdot)$ has absolute value bounded by $\frac{1}{2}$. Note also that the input to the square root is bounded by

$$\frac{\sigma(\boldsymbol{W}_{ij}^\star s)\,\text{sign}(\alpha)}{2W\alpha} \leq \frac{1}{2W|\alpha|} \leq \frac{1}{2W\mu} \tag{25}$$

Note further that $\alpha$ is known as an estimate $\hat{\alpha} := \frac{\widehat{\boldsymbol{v}}_j}{\widehat{\boldsymbol{v}}_i}$ up to error $\Delta_\alpha$ with $|\Delta_\alpha| = \frac{4\epsilon_v}{\mu^2}$:

$$|\Delta_\alpha| := |\hat{\alpha} - \alpha| = \left|\frac{\boldsymbol{v}_j^\star + \Delta_j}{\boldsymbol{v}_i^\star + \Delta_i} - \frac{\boldsymbol{v}_j^\star}{\boldsymbol{v}_i^\star}\right| = \left|\frac{\boldsymbol{v}_i^\star(\boldsymbol{v}_j^\star + \Delta_j) - \boldsymbol{v}_j^\star(\boldsymbol{v}_i^\star + \Delta_i)}{\boldsymbol{v}_i^\star(\boldsymbol{v}_i^\star + \Delta_i)}\right| = \left|\frac{\boldsymbol{v}_i^\star\Delta_j + \boldsymbol{v}_j^\star\Delta_i}{\boldsymbol{v}_i^\star(\boldsymbol{v}_i^\star + \Delta_i)}\right| \leq \frac{|\boldsymbol{v}_i^\star|\cdot|\Delta_j| + |\boldsymbol{v}_j^\star|\cdot|\Delta_i|}{|\boldsymbol{v}_i^\star|\cdot|\boldsymbol{v}_i^\star + \Delta_i|}$$

$$\leq \frac{2\epsilon_v}{\mu(\mu - \epsilon_v)} \leq \frac{4\epsilon_v}{\mu^2}$$

where $\Delta := \widehat{\boldsymbol{v}} - \boldsymbol{v}^\star$, and $\epsilon_v$ as defined in the theorem statement. Here, note that the argument used the sign of $\alpha$ earlier to choose $s = -\text{sign}(\alpha)/(2W)$; this is valid because $\text{sign}(\hat{\alpha}) = \text{sign}(\alpha)$, which follows from combining $|\alpha| \geq \mu$ with the bound on $|\hat{\alpha} - \alpha|$.

By construction, $\hat{a}, a^\star \in [0, \sqrt{\frac{1}{2W\mu}}]$. We then obtain, with $\Delta_a := \hat{a} - a^\star$ ($|\Delta_a| \leq \tau$):

$$\begin{aligned}
E := \left|\hat{a}^2 - (a^\star)^2\right| &= \left|(a^\star + \Delta_a)^2 - (a^\star)^2\right| \\
&= \left|(a^\star)^2 + 2\Delta_a a^\star + \Delta_a^2 - (a^\star)^2\right| \\
&= \left|2\Delta_a a^\star + \Delta_a^2\right| \\
&\leq 2\tau\sqrt{\frac{1}{2W\mu}} + \tau^2 \leq \tau\sqrt{\frac{2}{W\mu}} + \tau\sqrt{\frac{1}{2W\mu}} \leq \tau\frac{2}{\sqrt{W\mu}}
\end{aligned}$$

where we used $\tau \leq \sqrt{\frac{1}{2W\mu}}$.

Now

$$\left(-\frac{\sigma(\boldsymbol{W}_{ij}^\star s)s}{\alpha} + E\right) \cdot \hat{\alpha} = \left(-\frac{\sigma(\boldsymbol{W}_{ij}^\star s)s}{\alpha} + E\right) \cdot (\alpha + \Delta_\alpha)$$

$$= -\sigma(\boldsymbol{W}_{ij}^\star s)s + \underbrace{-\frac{\sigma(\boldsymbol{W}_{ij}^\star s)s}{\alpha}\Delta_\alpha + E\alpha + E\Delta_\alpha}_{\text{Estimation Error}}$$

where the estimation error is bounded by

$$\leq \underbrace{\left|\frac{\sigma(\boldsymbol{W}_{ij}^\star s)s}{\alpha}\right|}_{\leq 1/(2W\mu)} \underbrace{|\Delta_\alpha|}_{\leq 4\epsilon_v/\mu^2} + \underbrace{|E\alpha|}_{\frac{2\tau}{\sqrt{W\mu\mu}}} + \underbrace{|E\Delta_\alpha|}_{\frac{8\tau\epsilon_v}{\sqrt{W\mu\mu^2}}} \leq \frac{2\epsilon_v}{W\mu^3} + \frac{2\tau}{\sqrt{W\mu\mu}} + \frac{8\tau\epsilon_v}{\sqrt{W\mu\mu^2}} \leq \frac{2\epsilon_v}{W\mu^3} + \frac{10\tau}{\mu^3} \tag{26}$$

where we used $\epsilon_v, \mu \leq 1$; $W \geq 2$. Take $L$ to be the Lipschitz constant of $\sigma^{-1}$ on $[\sigma(-0.5), \sigma(0.5)]$; we have $4 < L < \frac{9}{2}$. We get the estimation error to be $\leq \frac{\eta_W}{4W^2 L}$, by setting

$$\tau := \frac{\eta_W \mu^3}{8LW^2} \cdot \frac{1}{50}$$

forcing the above estimation error to satisfy:

$$\begin{aligned}
(26) &\leq \frac{2}{W\mu^3}\frac{\mu^3\epsilon_W}{40Wd} + \frac{10}{\mu^3}\frac{\eta_W\mu^3}{8LW^2}\frac{1}{50}\\
&= \frac{\epsilon_W}{20W^2 d} + \frac{10}{8}\frac{\eta_W}{LW^2}\frac{1}{50}\\
&\leq \frac{\epsilon_W}{20dW^2} + \frac{10}{8}\frac{\eta_W}{W^2 L}\frac{1}{50}\\
&\leq \frac{\eta_W}{20W^2} + \frac{10}{8}\frac{\eta_W}{W^2 L}\frac{1}{50}\\
&\leq \frac{\eta_W}{4W^2 L}
\end{aligned}$$

given the choice $\epsilon_v = \frac{\mu^3\epsilon_W}{40Wd} = \frac{\mu^3\eta_W}{40W} \leq \frac{\mu^3\eta_W}{8LW}$ in the theorem statement. Now we have an estimate of

$$-\sigma(\boldsymbol{W}_{ij}s)s = \sigma(\boldsymbol{W}_{ij}s)\frac{\text{sign}(\alpha)}{2W} \tag{27}$$

with error bounded by $\frac{\eta_W}{4W^2 L}$. Multiplying by $2W$, we have an estimate of

$$\sigma(\boldsymbol{W}_{ij}s) \tag{28}$$

with error bounded by $\frac{\eta_W}{2WL}$.

Truncating to $[\sigma(-0.5), \sigma(0.5)]$ if needed (note that $|\boldsymbol{W}_{ij}s| \leq \frac{1}{2}$), we then get an estimate of $\boldsymbol{W}_{ij}s$ with error bounded as $\frac{\eta_W}{2W}$.

Multiplying by $s^{-1}$ gets us to an estimate of $\boldsymbol{W}_{ij}$ with error bounded as $\eta_W = \epsilon_W/d$, with (combining the expression for the number of queries in bisection with our expression for $\tau$)

$$\mathcal{O}\left(\log\frac{a_{\max}}{\tau}\right) = \mathcal{O}\left(\log\left(\frac{50 \cdot 8L}{\sqrt{2}} \cdot \frac{W^{3/2}}{\eta_W \mu^{7/2}}\right)\right) = \mathcal{O}\left(\log\left(\frac{50 \cdot 8 \cdot 9}{\sqrt{22}} \cdot \frac{W^{3/2}}{\eta_W \mu^{7/2}}\right)\right).$$

Repeating over all $(i, j) \in [d]^2$ gives the claimed

$$d^2 \log \left( C \cdot \frac{W^{3/2} d}{\epsilon_W \mu^{7/2}} \right),$$

term via a union bound, with an absolute constant $C$.

Overall, the estimator is

$$\widehat{\boldsymbol{W}}_{ij} := \frac{1}{s} \sigma^{-1} \Big( \mathrm{clip}_{\sigma(-1/2), \sigma(1/2)} \big( -\hat{a}^2 \hat{\alpha}/s \big) \Big),$$

where $\hat{a}$ is the bisection output and $s = -\mathrm{sign}(\hat{\alpha})/(2W)$.

## C. Multi-head Attention

For positive integers $H$ and $d_h$, let $d = H d_h$. We will have $H$ parameter matrices $\boldsymbol{W}^{(1)}, \ldots, \boldsymbol{W}^{(H)} \in \mathbb{R}^{d \times d}$ with rank $\leq d_h$ where $\boldsymbol{W}^{(h)} = (\boldsymbol{K}^{(h)})^\top \boldsymbol{Q}^{(h)}$ similar to the single head definition. We will have $H$ value projection matrices $\boldsymbol{V}^{(1)}, \ldots, \boldsymbol{V}^{(H)} \in \mathbb{R}^{d_h \times d}$ and one output projection vector $\boldsymbol{w}_o \in \mathbb{R}^{d \times 1}$. The $h$th attention weight will then be computed by

$$\boldsymbol{\alpha}^{(h)}(X, \boldsymbol{W}^{(h)}) = \mathrm{softmax}(\boldsymbol{x}_1^\top \boldsymbol{W}^{(h)} \boldsymbol{x}_N, \ldots, \boldsymbol{x}_N^\top \boldsymbol{W}^{(h)} \boldsymbol{x}_N) \in \Delta^{N-1}$$

and the output of the $h$th head will be,

$$\boldsymbol{a}^{(h)} = \boldsymbol{\alpha}^{(h)}(X, \boldsymbol{W}^{(h)})^\top (X(\boldsymbol{V}^{(h)})^\top) \in \mathbb{R}^{d_h}.$$

Following the standard definition, the outputs are then concatenated to compute the final output,

$$\boldsymbol{a} = [\boldsymbol{a}^{(1)}, \ldots, \boldsymbol{a}^{(H)}] \in \mathbb{R}^{d \times 1} \quad f_\theta^H(X) = \boldsymbol{a}^\top \boldsymbol{w}_o \in \mathbb{R},$$

where $\theta$ has all the parameter matrices $\boldsymbol{W}^{(h)}, \boldsymbol{V}^{(h)}$ for all $h$ and the vector $\boldsymbol{w}_o$. Note that we can merge the value matrices and output projection matrices in the following way. Partition $\boldsymbol{w}_o$ in $H$ equal and contiguous parts $\boldsymbol{w}_o = [\boldsymbol{w}_o^{(1)}, \ldots, \boldsymbol{w}_o^{(H)}]$ where $\boldsymbol{w}_o^{(h)} \in \mathbb{R}^{d_h}$. Then for vectors $\boldsymbol{v}^{(h)} = (\boldsymbol{V}^{(h)})^\top \boldsymbol{w}_o^{(h)} \in \mathbb{R}^{d \times 1}$, we can rewrite,

$$f_\theta^H(X) = \sum_{h=1}^H \boldsymbol{\alpha}^{(h)}(X, \boldsymbol{W}^{(h)})^\top (X(\boldsymbol{V}^{(h)})^\top) \boldsymbol{w}_o^{(h)} = \sum_{h=1}^H \boldsymbol{\alpha}^{(h)}(X, \boldsymbol{W}^{(h)})^\top (X \boldsymbol{v}^{(h)}).$$

Thus, the multi-head attention model can be seen as a sum of $H$ single-head attention models with their own parameters $(\boldsymbol{W}^{(h)}, \boldsymbol{v}^{(h)})$.

### C.1. Identifiability under Orthogonal Subspace Assumption

We show that the following orthogonal subspace assumption on the parameters $(W, v)$ suffices to make $f_{W, v}^H$ identifiable from queries.

**Assumption C.1.** Let the parameters of $h^{\mathrm{th}}$ head $\boldsymbol{W}^{(h)}$ and $\boldsymbol{v}^{(h)}$ lie in the subspace $U_h \subset \mathbb{R}^d$ and let $U_1, \ldots, U_H \subset \mathbb{R}^d$ be pairwise orthogonal subspaces with direct sum $\mathbb{R}^d = \bigoplus_{h=1}^H U_h$ and dimensions $d_h := \dim(U_h)$ (so $\sum_{h=1}^H d_h = d$). Let $\boldsymbol{P}_h \in \mathbb{R}^{d \times d}$ be the orthogonal projector onto $U_h$ ($\boldsymbol{P}_h^\top = \boldsymbol{P}_h$, $\boldsymbol{P}_h^2 = \boldsymbol{P}_h$, and $\boldsymbol{P}_g \boldsymbol{P}_h = \boldsymbol{0}$ for $g \neq h$). Assume for each head $h$ that

$$\boldsymbol{v}^{(h)} \in U_h \setminus \{\boldsymbol{0}\} \subset \mathbb{R}^d, \qquad \boldsymbol{W}^{(h)} \in \mathbb{R}^{d \times d} \text{ satisfies } \boldsymbol{P}_g \boldsymbol{W}^{(h)} = \boldsymbol{W}^{(h)} \boldsymbol{P}_g = \boldsymbol{0} \text{ for all } g \neq h,$$

equivalently, $\boldsymbol{W}^{(h)} = \boldsymbol{P}_h \boldsymbol{B}_h \boldsymbol{P}_h$ for some $\boldsymbol{B}_h \in \mathbb{R}^{d \times d}$ with $\mathrm{range}(\boldsymbol{B}_h) \subseteq U_h$.

**Proposition C.2** (Identifiability up to permutation). *Under the orthogonal subspace assumption (assumption C.1), the multi-head attention class is identifiable from queries, uniquely up to a permutation of the heads.*

*Proof.* First note that, for any $h \in [H]$, if we have $X \in \mathbb{R}^{N \times d}$ such that all rows of $X$ are in $U_h$, then $X\boldsymbol{v}^{(g)} = \boldsymbol{0}$ for all $g \neq h$. Thus, only a single head contributes to the output of the multi-head attention model,

$$f_{W,v}^H(X) = \sum_{g=1}^H \boldsymbol{\alpha}^{(g)}(X, \boldsymbol{W}^{(g)})^\top (X\boldsymbol{v}^{(g)}) = \boldsymbol{\alpha}^{(h)}(X, \boldsymbol{W}^{(h)})^\top (X\boldsymbol{v}^{(h)}).$$

Thus, on inputs supported in $U_h$, the multi-head model reduces exactly to the single-head model with parameters $(\boldsymbol{W}^{(h)}, \boldsymbol{v}^{(h)})$.

Let $\{\mathbf{u}_1^{(h)}, \ldots, \mathbf{u}_{d_h}^{(h)}\}$ be an orthonormal basis of $U_h$ and write $\boldsymbol{E}_h := [\mathbf{u}_1^{(h)} \cdots \mathbf{u}_{d_h}^{(h)}] \in \mathbb{R}^{d \times d_h}$, so that $\boldsymbol{P}_h = \boldsymbol{E}_h \boldsymbol{E}_h^\top$ and every $\boldsymbol{x} \in U_h$ has a unique coordinate vector $\boldsymbol{z} \in \mathbb{R}^{d_h}$ with $\boldsymbol{x} = \boldsymbol{E}_h \boldsymbol{z}$. For any $X$ whose rows lie in $U_h$ there exists $\boldsymbol{Z} \in \mathbb{R}^{N \times d_h}$ with $X = \boldsymbol{Z}\boldsymbol{E}_h^\top$ (the rows of $\boldsymbol{Z}$ are the coordinate rows). Define the reduced parameters

$$\widetilde{\boldsymbol{v}}^{(h)} := \boldsymbol{E}_h^\top \boldsymbol{v}^{(h)} \in \mathbb{R}^{d_h}, \qquad \widetilde{\boldsymbol{W}}^{(h)} := \boldsymbol{E}_h^\top \boldsymbol{W}^{(h)} \boldsymbol{E}_h \in \mathbb{R}^{d_h \times d_h}.$$

Using $\boldsymbol{W}^{(h)} = \boldsymbol{P}_h \boldsymbol{B}_h \boldsymbol{P}_h$ and $\boldsymbol{P}_h = \boldsymbol{E}_h \boldsymbol{E}_h^\top$, we have for all coordinate rows $\boldsymbol{z}_i, \boldsymbol{z}_N \in \mathbb{R}^{d_h}$,

$$\boldsymbol{x}_i^\top \boldsymbol{W}^{(h)} \boldsymbol{x}_N = (\boldsymbol{E}_h \boldsymbol{z}_i)^\top \boldsymbol{W}^{(h)} (\boldsymbol{E}_h \boldsymbol{z}_N) = \boldsymbol{z}_i^\top (\boldsymbol{E}_h^\top \boldsymbol{W}^{(h)} \boldsymbol{E}_h) \boldsymbol{z}_N = \boldsymbol{z}_i^\top \widetilde{\boldsymbol{W}}^{(h)} \boldsymbol{z}_N,$$

and $X\boldsymbol{v}^{(h)} = \boldsymbol{Z}\,\boldsymbol{E}_h^\top \boldsymbol{v}^{(h)} = \boldsymbol{Z}\,\widetilde{\boldsymbol{v}}^{(h)}$. Hence for such $X$,

$$f_{W,v}^H(X) = \boldsymbol{\alpha}(\boldsymbol{Z}, \widetilde{\boldsymbol{W}}^{(h)})^\top (\boldsymbol{Z}\,\widetilde{\boldsymbol{v}}^{(h)}),$$

which is precisely the single-head attention map in dimension $d_h$ with parameters $(\widetilde{\boldsymbol{W}}^{(h)}, \widetilde{\boldsymbol{v}}^{(h)})$ evaluated on input $\boldsymbol{Z}$.

*Recovery of $\widetilde{\boldsymbol{v}}^{(h)}$.* Query the oracle with one-row inputs of the form $X = [(\boldsymbol{E}_h \boldsymbol{e}_i)^\top]$, $i = 1, \ldots, d_h$, where $\{\boldsymbol{e}_i\}$ is the standard basis of $\mathbb{R}^{d_h}$. Since $N = 1$ makes the softmax weight 1, each response is $y = \langle \boldsymbol{E}_h \boldsymbol{e}_i, \boldsymbol{v}^{(h)} \rangle = \langle \boldsymbol{e}_i, \boldsymbol{E}_h^\top \boldsymbol{v}^{(h)} \rangle = \widetilde{\boldsymbol{v}}_i^{(h)}$. Thus $\widetilde{\boldsymbol{v}}^{(h)}$ is recovered from $d_h$ queries, and then $\boldsymbol{v}^{(h)} = \boldsymbol{E}_h \widetilde{\boldsymbol{v}}^{(h)}$.

*Recovery of $\widetilde{\boldsymbol{W}}^{(h)}$.* Fix $j \in [d_h]$ and set $\widetilde{\mathbf{w}}_j^{(h)} := \widetilde{\boldsymbol{W}}^{(h)} \boldsymbol{e}_j$ (the $j$-th column). For any probe $\widetilde{\boldsymbol{u}} \in \mathbb{R}^{d_h}$, query the two-row input

$$X = \begin{bmatrix} (\boldsymbol{E}_h \widetilde{\boldsymbol{u}})^\top \\ (\boldsymbol{E}_h \boldsymbol{e}_j)^\top \end{bmatrix} = \begin{bmatrix} \boldsymbol{x}_1^\top \\ \boldsymbol{x}_2^\top \end{bmatrix}, \qquad \boldsymbol{x}_1, \boldsymbol{x}_2 \in U_h.$$

As above, only head $h$ contributes and in the reduced coordinates the observed label is

$$y = \alpha(\widetilde{\boldsymbol{u}}; j)(\widetilde{\boldsymbol{u}}^\top \widetilde{\boldsymbol{v}}^{(h)}) + (1 - \alpha(\widetilde{\boldsymbol{u}}; j))\widetilde{\boldsymbol{v}}_j^{(h)}, \qquad \alpha(\widetilde{\boldsymbol{u}}; j) = \sigma((\widetilde{\boldsymbol{u}} - \boldsymbol{e}_j)^\top \widetilde{\mathbf{w}}_j^{(h)}).$$

Whenever $\widetilde{\boldsymbol{u}}^\top \widetilde{\boldsymbol{v}}^{(h)} \neq \widetilde{\boldsymbol{v}}_j^{(h)}$ we can read off

$$\alpha(\widetilde{\boldsymbol{u}}; j) = \frac{y - \widetilde{\boldsymbol{v}}_j^{(h)}}{\widetilde{\boldsymbol{u}}^\top \widetilde{\boldsymbol{v}}^{(h)} - \widetilde{\boldsymbol{v}}_j^{(h)}} \in (0, 1), \qquad \sigma^{-1}(\alpha(\widetilde{\boldsymbol{u}}; j)) = (\widetilde{\boldsymbol{u}} - \boldsymbol{e}_j)^\top \widetilde{\mathbf{w}}_j^{(h)}.$$

Thus each probe $\widetilde{\boldsymbol{u}}$ yields a linear equation in the unknown $\widetilde{\mathbf{w}}_j^{(h)}$. Choose $d_h$ probes $\widetilde{\boldsymbol{u}}_1, \ldots, \widetilde{\boldsymbol{u}}_{d_h} \in \mathbb{R}^{d_h}$ so that (i) $\widetilde{\boldsymbol{u}}_\ell^\top \widetilde{\boldsymbol{v}}^{(h)} \neq \widetilde{\boldsymbol{v}}_j^{(h)}$ for all $\ell$, and (ii) the differences $\widetilde{\boldsymbol{z}}_\ell := \widetilde{\boldsymbol{u}}_\ell - \boldsymbol{e}_j$ are linearly independent. (For instance, use the deterministic construction from the single-head case inside $\mathbb{R}^{d_h}$: pick any $p$ with $\widetilde{\boldsymbol{v}}_p^{(h)} \neq 0$ and set $\widetilde{\boldsymbol{u}}_\ell = \boldsymbol{e}_j + \boldsymbol{e}_\ell$ if $\widetilde{\boldsymbol{v}}_\ell^{(h)} \neq 0$, else $\widetilde{\boldsymbol{u}}_\ell = \boldsymbol{e}_j + \boldsymbol{e}_\ell + \boldsymbol{e}_p$.) Stacking the equations gives

$$\underbrace{\begin{bmatrix} (\widetilde{\boldsymbol{z}}_1)^\top \\ \vdots \\ (\widetilde{\boldsymbol{z}}_{d_h})^\top \end{bmatrix}}_{\widetilde{\boldsymbol{Z}} \in \mathbb{R}^{d_h \times d_h}} \widetilde{\mathbf{w}}_j^{(h)} = \underbrace{\begin{bmatrix} \sigma^{-1}(\alpha(\widetilde{\boldsymbol{u}}_1; j)) \\ \vdots \\ \sigma^{-1}(\alpha(\widetilde{\boldsymbol{u}}_{d_h}; j)) \end{bmatrix}}_{\widetilde{t} \in \mathbb{R}^{d_h}}, \qquad \alpha(\widetilde{\boldsymbol{u}}_\ell; j) = \frac{y_\ell - \widetilde{\boldsymbol{v}}_j^{(h)}}{\widetilde{\boldsymbol{u}}_\ell^\top \widetilde{\boldsymbol{v}}^{(h)} - \widetilde{\boldsymbol{v}}_j^{(h)}}.$$

By construction $\widetilde{Z}$ is invertible, so $\widetilde{\mathbf{w}}_j^{(h)} = \widetilde{Z}^{-1}\widetilde{t}$ is uniquely determined. Repeating over $j = 1, \ldots, d_h$ recovers $\widetilde{W}^{(h)}$; lifting back to $\mathbb{R}^d$ gives

$$W^{(h)} = E_h\, \widetilde{W}^{(h)}\, E_h^{\top}, \qquad v^{(h)} = E_h\, \widetilde{v}^{(h)}.$$

Because the heads decouple on subspace-restricted queries, the above procedure recovers $(W^{(h)}, v^{(h)})$ for each $h$ independently. If two parameter tuples in the class induce the same function on all $X$, then restricting to each $U_h$ and invoking the single-head identifiability gives equality of the corresponding $(W^{(h)}, v^{(h)})$ in that subspace. If the labels of the subspaces are not fixed, the parameters are recovered uniquely up to a permutation of the heads. This proves the proposition. $\qquad\square$

**Conjecture on Learning with Queries.** The above result indicates that under the orthogonal subspace assumption, if the learner knows the basis of each subspace, then it can essentially use the single-head approach to recover the parameters up to some permutation. The key challenge then lies in identifying the basis of each subspace.

We conjecture that the multi-head attention model could be learnable with access to value queries or potentially Hessian queries. We describe a potential direction to proving it in this section and discuss current obstacles.

One can make use of value queries to obtain the Hessian of a function (approximately) via standard finite differencing methods and obtain the Hessian for any input up to some additive tolerance. For the rest of the discussion, we assume that the learner has access to the true exact Hessian.

Take inputs of the form $X = \begin{bmatrix} (z + \mathbf{e}_j)^{\top} \\ \mathbf{e}_j^{\top} \end{bmatrix}$ where $z$ is an input probe vector. Analogous to the single-head case, the multi-head attention model can be rewritten as,

$$f(X) = \sum_{h=1}^{H} \left( \sigma(z^{\top}\mathbf{w}_j^{(h)})(z^{\top}v^{(h)}) + \mathbf{e}_j^{\top}v^{(h)} \right).$$

However, unlike the single-head case, we cannot take $\sigma^{-1}$ and solve it directly. Rewrite it and define the function $F(z) : \mathbb{R}^d \to \mathbb{R}$,

$$F(z) = f(X) - \mathbf{e}_j^{\top}\sum_{h=1}^{H} v^{(h)} = \sum_{h=1}^{H} \sigma(z^{\top}\mathbf{w}_j^{(h)})(z^{\top}v^{(h)}).$$

The Hessian $\nabla_z^2 F(z)$ has the following form,

$$\nabla_z^2 F(z) = \sum_{h=1}^{H} \left( (z^{\top}v^{(h)})\sigma''(z^{\top}\mathbf{w}_j^{(h)})\mathbf{w}_j^{(h)}(\mathbf{w}_j^{(h)})^{\top} + \sigma'(z^{\top}\mathbf{w}_j^{(h)})(\mathbf{w}_j^{(h)}(v^{(h)})^{\top} + v^{(h)}(\mathbf{w}_j^{(h)})^{\top}) \right).$$

We can write it in matrix form in the following way,

$$U^{(j)} = \begin{bmatrix} \mathbf{w}_j^{(1)} & \cdots & \mathbf{w}_j^{(H)} \end{bmatrix} \in \mathbb{R}^{d\times H}, \quad V = \begin{bmatrix} v^{(1)} & \cdots & v^{(H)} \end{bmatrix} \in \mathbb{R}^{d\times H}, \quad s = (U^{(j)})^{\top}z \in \mathbb{R}^H, \quad t = V^{\top}z \in \mathbb{R}^H.$$

Write $\sigma, \sigma', \sigma''$ elementwise on vectors. Define

$$D_1 = \mathrm{Diag}\big(\sigma'(s)\big), \qquad D_2 = \mathrm{Diag}\big(\sigma''(s) \odot t\big),$$

where $\odot$ is the Hadamard or elementwise product.

$$\nabla_z^2 F(z) = U^{(j)}D_2(U^{(j)})^{\top} + U^{(j)}D_1V^{\top} + \big(U^{(j)}D_1V^{\top}\big)^{\top}.$$

At the zero vector, we get the Hessian matrix with the following form,

$$4\nabla_{\boldsymbol{z}}^2 F(\mathbf{0}) = U^{(j)}V^\top + V(U^{(j)})^\top.$$

One approach is to try and leverage the structure of the Hessian to identify the basis of each subspace. If we have access to the Hessian, the task reduces to the following problem. For integers $d$ and $d_h < d/2$ such that $H = d/d_h$ is a positive integer, there are $H$ subspaces $S_1, \dots, S_H$ such that $S_i \perp S_j$ for all $i \neq j$ (Assumption C.1). So we have $H$ unknown spaces which are orthogonal to each other. There are unknown matrices $\boldsymbol{W}^{(1)}, \dots, \boldsymbol{W}^{(H)} \in \mathbb{R}^{d \times d}$ and vectors $\boldsymbol{v}_1, \dots, \boldsymbol{v}_H \in \mathbb{R}^d$ with the property that the columns of $\boldsymbol{W}^{(h)}$ and vector $\boldsymbol{v}_h$ are in the subspace $S_h$. Thus, the columns of any matrix $\boldsymbol{W}^{(i)}$ is orthogonal to the columns of $\boldsymbol{W}^{(j)}$ and $\boldsymbol{v}_j$ for all $i \neq j$.

For $j \in [d]$, define the matrix $U^{(j)} = [\boldsymbol{W}_j^{(1)}, \dots, \boldsymbol{W}_j^{(H)}]$ and $V = [\boldsymbol{v}_1, \dots, \boldsymbol{v}_H]$. We have access to $d$ matrices $A_1, \dots A_d$ where for $j \in [d]$

$$A^{(j)} = U^{(j)}V^\top + V(U^{(j)})^\top = \sum_{h=1}^H \big(\boldsymbol{u}_h^{(j)}\boldsymbol{v}_h^\top + \boldsymbol{v}_h(\boldsymbol{u}_h^{(j)})^\top\big)$$

Given matrices $\boldsymbol{A}^{(1)}, \dots, \boldsymbol{A}^{(d)}$, we want to identify the basis $\tilde{\mathbf{e}}_1, \dots, \tilde{\mathbf{e}}_d$ of the subspaces $S_1, \dots, S_H$ where $S_h$ has the orthonormal basis $\tilde{\mathbf{e}}_{(h-1)d_h+1}, \dots, \tilde{\mathbf{e}}_{hd_h}$. Note that each subspace can have many possible basis vectors, but we need to identify any one of them.

A couple of useful properties about the problem are the following. First, note that for any vector $\boldsymbol{x} \in S_i$, the vector $\boldsymbol{A}^{(j)}\boldsymbol{x} \in S_i$ and hence, each matrix $\boldsymbol{A}^{(j)}$ for $j \in [d]$ is invariant in each of the subspaces $S_1, \dots, S_H$. Secondly, if we use a projection matrix $\boldsymbol{E}$ using the orthogonal basis of the subspaces $\tilde{\mathbf{e}}_1, \dots, \tilde{\mathbf{e}}_d$ instead of the standard basis vectors, then matrices $\boldsymbol{E}^\top \boldsymbol{A}^{(1)}\boldsymbol{E}, \dots, \boldsymbol{E}^\top \boldsymbol{A}^{(d)}\boldsymbol{E}$ will all be simultaneously block-diagonal with blocks of sizes at most $d_h \times d_h$. Thus, a potential approach to extract the basis from the Hessian could be to use finest simultaneous block-diagonalisation algorithms (Murota et al., 2007; 2010). Given matrices $\boldsymbol{A}^{(1)}, \dots, \boldsymbol{A}^{(d)}$, a finest simultaneous block-diagonalisation (FSBD) algorithm produces a projection matrix $\boldsymbol{P}$ such that,

$$\boldsymbol{P}^\top \boldsymbol{A}^{(j)}\boldsymbol{P} \text{ is block-diagonal for all } j \in [d].$$

There are a few reasons why this does not establish any provable guarantees yet. Firstly, we can only hope to obtain the Hessian approximately via value queries. It is unclear whether block-diagonalisation can be guaranteed in that case and this approach could fail. Noise-robust versions of such algorithms (Maehara & Murota, 2011) require some separations in the eigenvalues of the target matrices, which would need some additional assumptions. Secondly, even in the case where we have access to exact Hessians, some degeneracy conditions could be needed since one could show that for blocks of size 1, there could be multiple sets of matrices $\boldsymbol{A}^{(1)}, \dots, \boldsymbol{A}^{(d)}$ which can be block-diagonalized using the same projection matrix when the eigenvalues collide. We believe that with mild degeneracy conditions and making use of probes other than zero vector $\nabla_{\boldsymbol{z}}^2 F(\mathbf{0})$, it could be possible to identify the basis from Hessian queries.

