# OpenReview forum: "Provably Learning Attention with Queries"
_ICML.cc/2026/Conference — ICML 2026 regular_

### Official Review · Reviewer_1aZK · 2026-02-20

**Soundness:** 3
**Presentation:** 4
**Significance:** 3
**Originality:** 3
**Overall Recommendation:** 5
**Confidence:** 4

**Summary:**

In this paper, the authors study the learning task whose objective is to recover the parameter matrices of a single self-attention layer. They mainly focus on four different cases: the self-attention layer, the self-attention layer combined with a fully-connected layer, the self-attention layer with low-rank key-query matrix, and the self-attention layer whose output is perturbed with a tolerance. Under these three cases, authors provide provable complexities of queries polynomial in the feature dimension $d$ and the low rank $r$. In addition, authors also discuss the multi-head attention settings, demonstrating that the parameter matrices are not identifiable, and hence difficult to recover.

**Compliance With Llm Reviewing Policy:**

Affirmed.

**Key Questions For Authors:**

My question is that compared to those works considering training [1, 2, 3, 4], the method proposed in this paper is superingly simple and effective. What are the main causes behind such effectiveness? See the details in the weaknesses part.

**Limitations:**

Yes

**Strengths And Weaknesses:**

**Strength:**
1. Compared to several theoretical works concerning the training of single-layer transformers [1, 2, 3, 4], which usually introduce the recovery of the parameter matrices as a sideproduct, this method proposed in this work is extremely effective, straightforward, and elegant. In the learning literature [1, 2, 3, 4], the problems usually rely on strong theoretical assumptions, like particular data distribution (Gaussian feature in [2, 3, 4] and Gaussian mixture in [1]), idealized population loss for training (considered in all [1, 2, 3, 4]), and certain requirements of feature dimension, sequence length for applying asymptotic analytical tools. In comparison, the method in this paper is simple but effective; it does not require any tedious optimization analysis, and the parameter matrices can be recovered by simply resolving linear equations. More importantly, [1, 2, 3, 4] all consider a strong regime that the attention weights are totally determined by position. In comparison, at least from my perspective, this paper can recover the key-query matrix with arbitrary input features, which may align better with the self-attention layer in real applications.

2. The writing of this paper is relatively good. The formulation of the problems, models, theorems, and proofs are quite clear and easy to follow. Personally, I really appreciate this straightforward writing style, which avoids introducing numerous unrelated complex concepts to wrap up the simple problem itself.

**Weaknesses**:
I believe that the techniques proposed in this paper are quite straightforward and effective. I do not see any major technical issues. In the following, I will only summarize some minor suggestions and questions, which may not present the weakness of this paper itself.

1. As I stated, the method proposed in this paper is quite different and effective compared to the classic literature consider the training of single-layer transformers. For me, such a technique is quite new, and I'm more familiar with the traditional optimization analysis. I would appreciate if the authors could reveal more details about the comparison between these two different paradigms. (I would definitely not penalize my evaluation even if the authors indicate any limitations of the current strategy.) For example, I feel that the current technique would fail for multiple-layer models, but the same problem also holds for optimization analysis in [1, 2, 3, 4].

2. The expression of Theorem 6.1 is a little unnatural. From my perspective, the tolerance should be defined as a part of the learning task itself. Therefore, a more natural statement might be for a given teacher model with specific output tolerance $\tau$, what are accuracy to recover the parameter matrices of this teacher model.

3. For completeness, I feel it would be better to give some intuitive explanation/critical steps of FFN and the compression sensing algorithm. (Although I understand that they are not the major focus of this paper.)

[1]. Jelassi et al. Vision transformers provably learn spatial structure.

[2]. Wang et al. Transformers provably learn sparse token selection while fully-connected nets cannot.

[3]. Zhang et al. Transformer learns optimal variable selection in group-sparse classification.

[4]. Zhang et al. Transformers trained via gradient descent can provably learn a class of teacher models.

---

> ### Author Rebuttal · Authors · 2026-03-30
>
> We thank the reviewer for their thoughtful feedback and time.
>
>
> > Q1. As I stated, the method proposed in this paper is quite different and effective compared to the classic literature consider the training of single-layer transformers. For me, such a technique is quite new, and I'm more familiar with the traditional optimization analysis. I would appreciate if the authors could reveal more details about the comparison between these two different paradigms. (I would definitely not penalize my evaluation even if the authors indicate any limitations of the current strategy.) For example, I feel that the current technique would fail for multiple-layer models, but the same problem also holds for optimization analysis in [1, 2, 3, 4].
>
> The methods for learning from queries and learning from examples are naturally quite different since their objectives are separate. In case of learning from examples, we typically have a loss along with a gradient-based algorithm (for deep learning models), hence the technical tools for analysis, such as optimisation, are natural. For query learning or parameter recovery, the objective is to typically figure out clever ways to construct queries and recover the target function exactly or approximately. So the analysis focuses more on exploiting the structure of the problem/model.
>
> The difficulty of the problem varies across models or classes of functions. As our results show, this particular problem of learning single-head attention admits a surprisingly simple algorithm. It is unclear whether such direct algorithms exist for multi-head attention under mild assumptions. The multi-layer model becomes even more complex given that there are feedforward networks involved in Transformers as well, so such straightforward approaches are unlikely to hold.
>
>
> Some of the techniques for query learning do generalise across model families. For instance, access to queries allows a learner to estimate gradients and Hessians by querying $f(x)$ and $f(x + \epsilon)$. Sometimes the gradients or Hessian at relevant inputs can help identify the weights. Such approaches have been taken in prior works on query learning ReLU FFNs (discussed in Sec. 2) and we explore such an approach for Multi-head attention under a somewhat strong assumption in Sec. C.
>
>
>
>
>
>
>
> > Q2. The expression of Theorem 6.1 is a little unnatural. From my perspective, the tolerance should be defined as a part of the learning task itself. Therefore, a more natural statement might be for a given teacher model with specific output tolerance $\tau$, what are accuracy to recover the parameter matrices of this teacher model.
>
> We agree that this is a subjective point and we decide to frame it in a certain way that follows closely to the classical statistical query learning model [1]. However, we agree that one might prefer that perspective. It can be naturally inferred from the existing result, and in the next version, we will discuss it clearly and explicitly specify the accuracy for fixed tolerance values.
>
>
> > Q3. For completeness, I feel it would be better to give some intuitive explanation/critical steps of FFN and the compression sensing algorithm. (Although I understand that they are not the major focus of this paper.)
>
> In the next version, we will discuss the high-level approach for learning FFNs in the discussion paragraph of Sec 4.1 and expand on the compressed sensing approach in Sec. A.1.
>
>
>
> [1] Efficient noise-tolerant learning from statistical queries. Kearns JACM 1998.

---

> > ### Author Rebuttal · Reviewer_1aZK · 2026-04-01
> >
> > I thank the authors for their detailed rebuttal! For the first question, to my knowledge, the current optimization analysis regarding transformers is also usually established on the recovery of the parameter matrices, and these works [1,2, 3, 4] can also cover the one-layer case. This comment might be unfair, but at least from my perspective, these works might be meaningless. The authors claim that these are caused by the discrepancy among the learning problems, but I still do not get the point. I would appreciate it if the authors could share more opinions regarding this point and I believe these discussions and comparisons would be valuable for the readers from the theory community.

---

> > > ### Author Response · Authors · 2026-04-04
> > >
> > > Thank you for your response.
> > >
> > > We will include a discussion in the paper comparing the two lines of work. The main distinction is that the two settings study different objectives and give the learner different powers. In [1-4], the learner receives labelled examples from a data distribution, and the goal is to show that gradient descent on a Transformer reaches a predictor with low error under that distribution. Those works therefore study passive learning and the behaviour of standard training algorithms.
> > >
> > > In contrast, our setting is a parameter recovery problem with query access. The learner is not given a data distribution, but can adaptively choose which inputs to query, with the explicit goal of extracting the target weights. This power is not available in the passive setting, where one cannot control the inputs and hence cannot isolate quantities in the same way. At the same time, our goal is also more stringent, since we aim to recover the parameters themselves rather than only achieve low error. For the problem of single-head attention, our results show that access to a query oracle leads to direct algorithms for parameter recovery, but the same may not hold for multi-head or multi-layer models. For some other unrelated class of functions like Conjunctions, Disjunctions, Singletons, etc, getting access to examples from the target distribution is more helpful, which allows efficient PAC-learning, but exact identification from a polynomial number of queries is a hard problem.

---

### Official Review · Reviewer_DyyV · 2026-02-26

**Soundness:** 3
**Presentation:** 3
**Significance:** 1
**Originality:** 2
**Overall Recommendation:** 3
**Confidence:** 4

**Summary:**

In this work, the authors study the problem of learning single-head attentions with queries.

First, they show that, by probing the entries of attention weights using appropriate input and query tokens, the weight matrix can be recovered with $O(d^2)$ queries, where $d$ is the embedding dimension.
In particular, this implies that if there exists a query-based algorithm that learns 2-layer networks, then by combining it with the above attention learning algorithm, we can efficiently learn a single-layer transformer.

Then, they show that, in the regime where the rank $r$ of the attention weight matrix is much smaller than the embedding dimension $d$,
the query complexity can be improved to $O(r d)$ using compressed sensing techniques.
They also study the learnability of single-head attentions when the query is noisy and show that the algorithm is robust under mild conditions.
Finally, they argue that, in general, the weights of multi-head attentions are not identifiable.

**Compliance With Llm Reviewing Policy:**

Affirmed.

**Final Justification:**

I would give this paper a borderline score if that were an option this year, because the theoretical analysis itself is solid, but I personally still don't think the setting is interesting for the reasons I listed in the initial review and the rebuttal acknowledgement. However, as I have said in the rebuttal acknowledgement, I understand that whether I find a problem interesting is partly subjective, and I am willing to defer to the opinions of the other reviewers and the AC.

As noted by the authors in their second rebuttal comment, it might be helpful to rephrase the discussion section to set the right expectations.

**Key Questions For Authors:**

See Weakness. In particular, can you clarify why learning a single-head attention with queries is an interesting task?

**Limitations:**

Yes.

**Strengths And Weaknesses:**

Overall, this is an okay paper.

The paper is well-written and easy-to-follow.
Though I did not check the proof details, the analysis looks believable.
For this specific task (learning single-head attentions/transformers), the authors provide a reasonably comprehensive analysis, including the query complexity in the general and low rank cases and the noise stability.

The main issue of this paper is that the task it studies is not interesting.
The authors argue that the results are "surprising" by comparing with query learning two-layer networks.
However, I do not think this is the right comparison, and the more appropriate analog here is query learning a single neuron.

For the task of learning attentions with queries, one can make "degree-2" queries by choosing the tokens $x_1, x_N$ appropriately to essentially force the model to return a monotonically transformed version of $W_{ij}$ (under a basis that may differ from the standard one).
This is closer to learning a single neuron, where the ground-truth weight is a vector, and we can make "degree-1" queries such as $\sigma( w \cdot e_i )$ to directly recover the entries of $w$.
Learning two-layer networks with queries is interesting only when there are more than one neuron, and I think the same is also true for learning attentions.

Regarding the multi-head setting, the non-identifiability result (Prop. 7.1) is almost trivial:. The authors simply choose the attention weights to be the same and vary the output weights.
The positive result (Prop. C.2) on recovering multi-head attentions requires the weights to lie in orthogonal subspaces, which more or less reduces the problem to the single-head case.

As a side note, it was shown in [1] that learning two-layer networks with queries is possible under fairly general conditions.
This work is very recent, so I do not think it is reasonable to expect the authors to be aware of it.
I mention it here as evidence suggesting that learning multi-head attentions with queries (the more interesting task) should also be feasible.

[1] Allen Liu. Provably Extracting the Features from a General Superposition. 2025

---

> ### Author Rebuttal · Authors · 2026-03-30
>
> We thank the reviewer for their thoughtful feedback and for noting that the paper provides a comprehensive analysis of the single-head case. We address the concerns below.
>
> **On the single-neuron analogy.** We do not believe the analogy to a single neuron is completely justified. First, the structure of the problem is different: the parameters consist of a matrix $W \in \mathbb{R}^{d \times d}$ and a vector $v \in \mathbb{R}^d$, not just a single vector. Second, consider composing our single-head attention model with a sigmoid, i.e., $\sigma(f_{W,v}(X))$. Since $\sigma$ is invertible, our extraction algorithm directly applies to it, and the model $\sigma(f_{W,v}(X))$ is a strict superset of the single-neuron model.  For length-1 queries, this model computes $\sigma(x^\top v)$, which is exactly the single-neuron problem and we say that recovering $v^*$ from queries is trivial (Phase (i) of Theorem 4.1). The goal is hence to figure out how $W$ can be recovered and an insight from our work is that with certain forms of length-2 queries we can reduce softmax to an invertible sigmoid and identify W. In this sense, what the reviewer describes as "degree-2" queries is closely related to the core mechanism underlying our recovery result, and we will clarify this intuition more explicitly in the paper. We do state in the paper that the resulting recovery rule is elementary; we view this as a positive aspect, since it shows that a model that appears significantly more difficult to learn from examples admits clean recovery with queries.
>
>
> **Questions beyond single-neuron learning.** We also believe the single-neuron analogy does not capture several parts of the paper that are specific to attention and have no direct counterparts in the single neuron setting. In particular, the low-rank regime is meaningful here because the attention parameter is matrix-valued, and this allows an $O(rd)$ recovery guarantee via rank-one measurements and matrix sensing. Likewise, the robustness analysis while handling the interaction between $W$ and $v$, and the membership-query extension address how the softmax-based recovery behaves under approximate/noisy oracle access. These are attention-specific aspects of the problem, and part of our contribution is to analyse them in a reasonably complete way for the single-head case.
>
> **Comparison with learning from examples.** For the single neuron model, efficient distribution-free PAC learning is possible and is classically known. For single-head attention, the situation is quite different. A body of work has sought to establish learning guarantees for specific problems and under various assumptions via a range of technical tools (See Sec. 2 and end of Sec. 4). Distribution-free PAC guarantees with minimal assumptions have not been established for the single-head attention model, despite considerable effort. While such algorithms and guarantees could certainly exist and the problem itself is different, given the current state of the literature, the learnability of single-head attention from examples appears substantially less understood than that of a single neuron. In that sense, one might not expect such a simple parameter-recovery algorithm from queries a priori.
>
>
>
> **On multi-head attention.** We agree with the reviewer that the multi-head problem is more interesting. Regarding Prop 7.1, we agree that the construction is simple but it does establish that algorithms for multi-head attention must impose some structural assumptions beyond those needed for the single-head case, which is useful for directing future work on the problem. We agree that developing algorithms for multi-head attention under mild structural assumptions is an important next step, and we hope the identifiability analysis and the directions outlined in Appendix C serve as useful starting points.
>
> Thanks for sharing the reference, and we will also discuss it in the next version.
>
> >  Q. Can you clarify why learning a single-head attention with queries is an interesting task?
>
> To understand the complexity of query learning Transformers, a natural starting point is the analysis of the simplest building block, which is single-head attention. With a similar motivation, a body of work (see Sec 2) has studied the learnability, optimisation dynamics, and other properties of single-head attention and even simpler models.
>
> As a first work on query-based learning for attention, we see part of the contribution as laying groundwork: a comprehensive analysis of the single-head case, the identifiability barrier for multi-head, and the approaches we outline (Hessian-based recovery via finite differencing, block-diagonalisation) that we hope will be useful to researchers working on the multi-head case and other variants of the problem.

---

> > ### Author Rebuttal · Reviewer_DyyV · 2026-04-01
> >
> > I thank the authors for the clarification.
> > Unfortunately, I am still not convinced that query learning a single-head attention is an interesting task, and will maintain my score (3).
> > However, I understand that whether I find a problem interesting is partly subjective, and I am willing to defer to the opinions of the other reviewers and the AC.
> >
> > **On the relation with query learning a single neuron.**
> > I agree that query learning a single-head attention is harder than query learning a single neuron, because of the nonlinear softmax and existence of the output layer.
> > However, query learning a single neuron is almost trivial, so being harder than that doesn't imply the task is interesting.
> > Also, when compared with query learning a two-layer network (the example the authors use in Related Work to justify this task), query learning a single-head attention is closer to learning a single neuron than learning a two-layer network, since after all, we can always write the attention part as $x_1^T W x_2 = (\mathrm{vec} W)^T(\mathrm{vec}(x_2 x_1^T))$ and directly probe the entries of $W$ by choosing $\mathrm{vec}(x_2 x_1^T)$ to be the basis vectors.
> > I understand that the proof is not this straightforward, but my impression is that this is the underlying idea.
> > For the low-rank results, I also don't think they are unique to this matrix problem, as one can similarly study recovery of a sparse weight vector (the analogue of low-rank matrices in the vector setting) using compressed sensing techniques such as Basis Pursuit.
> >
> > **Comparison with learning from examples.**
> > Again, I don't think the comparison is fair here, and this is also where I disagree with reviewer 1aZK.
> > Query model is much stronger than random examples and statistical query (SQ) models.
> > For example, the SQ lower bound for noisy k-parity is $d^{\Omega(k)}$, while this task can be easily solved with $\tilde{O}(d)$ queries.
> > Given the extra power of the query model, I would naturally expect much stronger theoretical guarantees.
> > For example, learning 2-layer networks (without assuming the weights are near-orthogonal) using random samples is generally believed to be hard because (the population of) it covers tensor decomposition, while query learning 2-layer networks is possible under fairly mild conditions.
> > I don't feel learning a single-head attention can provide separation close to this.

---

> > > ### Author Response · Authors · 2026-04-04
> > >
> > > Thank you for your response.
> > >
> > >
> > > **Re: Query learning single neuron and ReLU MLPs**
> > >
> > > We largely agree on some of the technical points you have made. We would like to elaborate on the comparison with ReLU MLPs. For $v \in \mathbb{R}^d$ and $W \in \mathbb{R}^{d \times d}$,  the ReLU MLP model has the form $f_{W, v}(x) = v^{\top} \operatorname{ReLU}(Wx)$. So on the surface, the single-head attention is similar in the sense that it has a matrix $W$, a vector $v$, and a different nonlinearity, which is softmax. Differently, the attention model takes a sequence of vectors as input rather than a single vector. We do not claim that any results on ReLU MLP have any implications for learning single-head attention. We mention it for two reasons: (i) Our result and algorithm show that the single-head attention query learning problem is much simpler despite the similarities. We do not claim that the sophistication of algorithms of learning ReLU MLPs implies any difficulty for the single-head attention problem, and that would actually be contradictory to our results. (ii) Second, the problem of query learning ReLU MLPs has received significant attention, whereas, as far as we are aware, similar work on query learning attention is scarce if not non-existent, despite the central role of attention in modern deep learning models. We agree that the current phrasing could give the wrong impression, and we will rephrase it in the next version to not overstate the comparison with ReLU MLPs.
> > >
> > >
> > >
> > > **On Comparison with learning from examples.**
> > >
> > >
> > >  It is not necessarily the case that learning from queries is a stronger model of learning than learning from examples. In fact, there are several classes of functions like Conjunctions, Disjunctions, Singleton, etc,  that are efficiently PAC-learnable from examples but cannot be identified with a polynomial number of queries. The Parity problem does have an elementary PAC-learning algorithm with $O(d)$ examples via Gaussian elimination, and is the classic example that separates PAC-learning and SQ-learning. The differences between learning from examples and query learning stem from a few things: (i) the objectives are different (target identification vs achieving low error over a distribution), (ii) Examples provide information about a target distribution, whereas learning from only queries is not guided by a distribution. That said, of course, in many cases, query learning can be easier than learning from examples, which is the case for the single-head attention problem and arguably other neural networks.
> > >
> > > Regardless, our intention is not to compare the complexity of learning in different models here. The discussion part is intended to provide context on what is known about related problems and to show where our results fit.
> > >
> > >
> > > We agree that it is not objectively surprising in the sense that it contradicts or goes against any prior results or implications. We will rephrase the discussion part and clarify that the points about ReLU MLPs and learning from examples are meant for additional context about related problems. And, given that context, it is somewhat surprising to us that this problem admits such a simple algorithm and not that there is anything inherently surprising about the result.

---

### Official Review · Reviewer_o7H2 · 2026-03-09

**Soundness:** 4
**Presentation:** 3
**Significance:** 2
**Originality:** 3
**Overall Recommendation:** 4
**Confidence:** 3

**Summary:**

This paper provides a theoretical analysis of using query based methods to uncover the weights of an attention-only transformer model. The authors first demonstrate that, given blackbox access to a single-head softmax attention layer, one can recover the weights of the attention heads using a polynomial number of queries. They then extend this analysis to the case where the model has a ReLU MLP for which a learner already exists and find that the attention weights are still recoverable in polynomial time. They then show that their algorithm can be done more efficiently when the attention heads are low rank and analyze the robustness of their method when a small amount of noise is injected into the model inputs. Finally, they show that multi-headed attention does not share the same polynomial time guarantees, and provide an initial discussion on how multi-headed attention might be approached.

**Compliance With Llm Reviewing Policy:**

Affirmed.

**Final Justification:**

I believe that this work provides interesting theoretical contributions, but it requires significant limitations in the attention module relative to real models, and it lacks a clear direction for future work. This makes its significance somewhat limited.

**Key Questions For Authors:**

1. This paper considers only single layer models. Do you have any initial insights into how these findings might translate into multi-layer (attention only) models?
2. Have you considered analyzing Grouped-Query Attention or similar alternatives to MHA? These may provide more theoretical guarantees than MHA as they have less flexibility in their model weights.

**Limitations:**

Yes

**Strengths And Weaknesses:**

__Soundness__

The theoretical claims in this paper are well supported by rigorous analysis and discussion. The authors also provide extensive discussion of limitations of their theory, such as the lack of MLP or multiple attention heads. These limitations seem reasonable given the complexity of transformer models and the relatively early stage of theoretical work in this field.

There are a few additional limitations that could have been acknowledged. This work requires direct access to the model inputs when many transformer based applications would only allow text or image inputs. Additionally, the paper only considers single-layer models.

Finally, the paper also may have been strengthened by a brief empirical study, even on synthetic or toy data, though I recognize that this may fall outside the primary theoretical scope of the work.

__Presentation__

The paper is generally well written, though the formatting could be improved. The paragraphs are quite dense and many equations and definitions are either fully or partially embedded within the text (rather than on separate lines or numbered) which makes it difficult to read at times. The paper would also benefit from clearly labeled subsections, as the bolded subsection headers again hurt readability.

Additionally, as your analysis requires many assumptions that do not hold within real-world models, a dedicated limitations or future work section would improve the credibility of your work and make it easier for future papers to address these issues. It currently is unclear how this study would extend to multi-layer models or multi-headed attention.

It seems possible that these formatting choices were made for the sake of space, as there is a lot of content to cover within the main paper. I might suggest moving some of the secondary studies to the appendix, such as the low-rank or robustness analyses, to allow the other sections to be more cleanly presented.

__Significance__

Transformer models see a significant amount of use today, especially via APIs, so understanding how these APIs might be exploited has a lot of practical application. With that said, many key assumptions of the paper do not reflect real-world models and the paper does not provide a clear path towards improving these gaps in the theory.

__Originality__

I am not very familiar with theoretical work regarding adversarial ML, but it seems like little theoretical work has been done on analyzing query based attacks on transformers. This paper provides a novel theoretical analysis and identifies how it differs from previous work, such as empirical methods or theoretical analyses on linear attention.

---

> ### Author Rebuttal · Authors · 2026-03-30
>
> We thank the reviewer for their thoughtful feedback and for appreciating the soundness of the paper.
>
>
> > Q. This paper considers only single layer models. Do you have any initial insights into how these findings might translate into multi-layer (attention only) models?
>
> We do not have any concrete results for multi-layer models. Our current hypothesis is that the parameters of multi-layer attention-only models may not be identifiable without any additional assumptions. Additionally, our current techniques, such as using short (length-1) queries, may partially extend to recover crucial information about the value matrices. However, recovering the attention matrices in deeper architectures appears more difficult and would likely require new ideas beyond our current framework. We view this as an important direction for future work. That said, our primary goal was to provide a clean and comprehensive understanding of the learnability of single-head attention models with queries and we hope some of the insights and approaches could be helpful for the multi-layer problem.
>
>
> > Q. Have you considered analyzing Grouped-Query Attention or similar alternatives to MHA? These may provide more theoretical guarantees than MHA as they have less flexibility in their model weights.
>
> We believe that the non-identifiability phenomenon we establish with query access extends to Grouped-Query Attention (GQA) when the number of groups exceeds one, as the construction in Proposition 7.1 can be adapted to this setting. For Multi-Query Attention (MQA), the non-identifiability issue remains less clear. While we agree that GQA and MQA impose additional structural constraints and reduce model flexibility, it remains unclear whether these constraints are sufficient to overcome the eigenvalue-collision issue highlighted in our approach (Appendix C).

---

> > ### Author Rebuttal · Reviewer_o7H2 · 2026-04-02
> >
> > Thank you for your response. I still believe that this work provides interesting theoretical contributions, but its inherent limitations and lack of clear future directions makes its significance somewhat limited. As such, I've elected to keep my score as it is.

---

### Official Review · Reviewer_kvZJ · 2026-03-12

**Soundness:** 3
**Presentation:** 4
**Significance:** 3
**Originality:** 4
**Overall Recommendation:** 5
**Confidence:** 3

**Summary:**

The submission examines whether single- and multi-head attention are identifiable using value queries to an oracle. Letting $d$ be the input dimension and $r$ the attention matrix rank, it proves that single-head attention is exactly identifiable in $O(d^2)$, and in $O(rd)$ when $r \ll d$. Further, the preprint proves identifiability under noisy oracle outputs. The proofs are by construction of the learning algorithms. Finally, the manuscript demonstrates that multi-head attention is not identifiable, as different sets of parameters can produce the same function.

**Compliance With Llm Reviewing Policy:**

Affirmed.

**Final Justification:**

The preprint initiates an original study of attention and transformer identifiability, adding a new perspective to transformer analysis, particularly regarding model security and safety. The presentation is clear and easy to follow, including the proofs.

I did not identify any major weaknesses, and my minor question was answered during the rebuttal.

**Key Questions For Authors:**

I give the submission a high score and only have a minor question to confirm my understanding:

Q1. In the proof of Proposition 7.1, would the multi-head attention be identifiable if the weights $\lambda_1, \ldots, \lambda_H$ were fixed?

**Limitations:**

yes

**Strengths And Weaknesses:**

**Strengths**

S1. The presentation is clear and easy to follow, including the proofs.

S2. Based on the Related Work section, the preprint initiates an original study of attention and transformer identifiability, providing the first results on their learnability from queries. These theoretical contributions add a new perspective to the growing body of work on transformer analysis, particularly regarding model security and safety.

**Weaknesses**

W1. I did not identify any major weaknesses. The significance could be improved by discussing multiple-layer transformers with a single-head attention or by a more in-depth analysis of conditions under which multi-head attention is identifiable, as noted in the submission.

---

> ### Author Rebuttal · Authors · 2026-03-30
>
> We thank the reviewer for their thoughtful feedback and time.
>
> > Q1.  In the proof of Proposition 7.1, would the multi-head attention be identifiable if the weights $\lambda_1, \ldots, \lambda_H$ were fixed?
>
> Not necessarily. We aimed to provide a simple proof to show that the parameter-function map is not injective. Even if $\lambda_1, \ldots, \lambda_H$ are fixed, it might be possible that different weight matrices for different heads could still lead to the same function. We do not have a direct answer for that at the moment but one needs to analyse the conditions required for identifiability.

---

> > ### Author Rebuttal · Reviewer_kvZJ · 2026-04-03
> >
> > I thank the authors for their response. My question was answered. I am keeping my positive score.

---

### Decision · Program_Chairs · 2026-04-30

**Decision:**

Accept (regular)

**Comment:**

The paper studies the problem of learning an "attention head" using queries. The results appear quite clean (and also relatively simple), because you are allowed to query basis vectors (slightly modified) and this lets them find the entries of the corresponding matrix. More work is needed for the extensions that are considered.

The reviewers are all quite positive about the paper and feel that the results, though not too surprising in hindsight, are interesting enough. I lean towards acceptance.